# Unlocking the Value of Text: Event-Driven Reasoning and Multi-Level Alignment for Time Series Forecasting

**Siyuan Wang, Peng Chen, Yihang Wang, Wanghui Qiu, Chenjuan Guo, Bin Yang, Yang Shu**[✉]
East China Normal University
{sywang,pchen,yhwang,onehui}@stu.ecnu.edu.cn,
{cjguo,byang,yshu}@dase.ecnu.edu.cn

## Abstract

Existing time series forecasting methods primarily rely on the numerical data itself. However, real-world time series exhibit complex patterns associated with multimodal information, making them difficult to predict with numerical data alone. While several multimodal time series forecasting methods have emerged, they either utilize text with limited supplementary information or focus merely on representation extraction, extracting minimal textual information for forecasting. To unlock the Value of Text, we propose VoT, a method with Event-driven Reasoning and Multi-level Alignment. Event-driven Reasoning combines the rich information in exogenous text with the powerful reasoning capabilities of LLMs for time series forecasting. To guide the LLMs in effective reasoning, we propose the Historical In-context Learning that retrieves and applies historical examples as in-context guidance. To maximize the utilization of text, we propose Multi-level Alignment. At the representation level, we utilize the Endogenous Text Alignment to integrate the endogenous text information with the time series. At the prediction level, we design the Adaptive Frequency Fusion to fuse the frequency components of event-driven prediction and numerical prediction to achieve complementary advantages. Experiments on real-world datasets across 10 domains demonstrate significant improvements over existing methods, validating the effectiveness of our approach in the utilization of text. The code is made available at https://github.com/decisionintelligence/VoT.

## 1 Introduction

Time series forecasting is a fundamental task in numerous domains, including financial market analysis, climate monitoring (Tian et al., 2025), and healthcare management, traffic awareness (Guo et al., 2014), and spatio-temporal tracking (Lu et al., 2011). Deep learning–based forecasting methods have achieved competitive performance (Nie et al., 2023; Chen et al., 2024; Qiu et al., 2025d; Wu et al., 2025; 2022) on this task. However, they solely rely on numerical time series data, limiting their ability to capture more complex patterns.

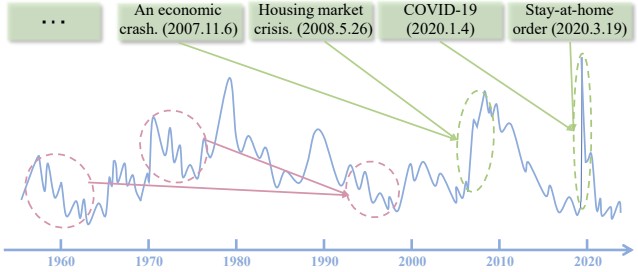

Figure 1: Unemployment rate time series (Liu et al., 2024a) (1970-2020). While certain patterns (pink) exhibit predictable temporal regularities, abrupt changes (green) driven by external events necessitate the integration of textual information to complement numerical forecasting.

Taking the unemployment rate data (Liu et al., 2024a) as an example, Figure 1 illustrates that certain abrupt changes in the time series are difficult to predict solely from historical numerical patterns. However, by incorporating textual information, the model may get crucial guidance that helps fore-

cast sudden shifts, such as the unemployment rate spikes triggered by the 2008 financial crisis and the COVID-19 pandemic in 2020. This demonstrates that text serves as a valuable complementary modality, enriching time series forecasting with event-driven guidance that is not easily inferred from numerical data alone. Moreover, textual information predominantly contributes to capturing event-related dynamics, but lacks descriptions of subtle fluctuations. However, this missing information can be supplemented through time series modeling. By jointly leveraging these complementary strengths, the model can better integrate the forecasting abilities of the two modalities, thereby breaking the limitations of single modality.

To explore the potential of textual information and effectively apply it in time series forecasting, there exist two main challenges. **The first challenge** is insufficient text utilization. Some approaches incorporate prompts based on *endogenous text* (Kowsher et al., 2025), such as statistical summaries or dataset-specific descriptions. While this text offers useful context, it largely overlaps with information already present in the time series. Consequently, they are unable to effectively model external drivers of temporal dynamics. On the other hand, some methods (Luo et al., 2023; Zhang et al., 2024) leverage LLMs to embed *exogenous text* (Liu et al., 2024a), which has richer textual sources, such as news and policy documents. These methods primarily focus on the representation-level fusion and are difficult to explore the deep semantic information, leaving the potential value of textual information untapped. **The second challenge** is effectively aligning two modalities to leverage their complementary strengths. While text provides crucial guidance for sudden shifts and event-related dynamics, time series modeling captures the subtle fluctuations and numerical trends that text cannot describe. However, the considerable modality gap between these modalities prevents existing methods from achieving effective cross-modal integration.

Based on these insights, we propose VoT, a novel multimodal time series forecasting method that leverages Event-driven Reasoning and Multi-level Alignment to effectively unlock the value of textual information. Our approach employs a dual-branch architecture to integrate both exogenous and endogenous text for comprehensive time series forecasting. *Event-driven Reasoning* is performed through our Event-Driven Prediction Branch, which employs a generative pipeline that includes template generation, summarization, and reasoning. This pipeline enables LLMs to extract more forecasting-related information from exogenous text. Additionally, we introduce Historical In-Context Learning (HIC), which provides error-informed guidance from historical samples for reasoning. *Multi-level alignment* is designed to fully integrate the predictive capabilities of time series and text. At the representation level, we introduce the numerical prediction branch with an Endogenous Text Alignment (ETA). The ETA first converts temporal statistics into textual descriptions and then uses decomposed pattern extraction and decomposed contrastive learning to achieve alignment between two modalities. At the prediction level, we introduce the Adaptive Frequency Fusion (AFF) that dynamically adjusts importance and integrates the frequency components of outputs from the event-driven prediction branch and the numerical prediction branch. Based on dynamic frequency fusion, the AFF enables domain-driven optimization to achieve complementary advantages and address the varying dependencies on textual and numerical information across different datasets. Specifically, we make the following contributions:

- We introduce an event-driven reasoning method to extract the forecasting-related information from exogenous text and obtain numerical predictions. It is enhanced by Historical In-Context Learning (HIC), which retrieves historical reasoning examples as prompts to provide error-informed guidance for reasoning. The method improves the reasoning ability of LLMs and unlocks the value of text.

- We propose a multi-level alignment approach. Specifically, we introduce the Endogenous Text Alignment (ETA) for representation-level alignment and the Adaptive Frequency Fusion (AFF) for prediction-level alignment. Through comprehensive alignment, we achieve complementary advantages across both modalities.

- We conduct extensive experiments on 10 real-world datasets from different domains and achieve state-of-the-art prediction accuracy. Moreover, we conduct thorough ablation and analysis experiments to demonstrate our effective utilization of text.

## 2 RELATED WORK

**Time Series Forecasting.** Time series forecasting has evolved from traditional statistical methods (Hyndman & Khandakar, 2008; Li et al., 2022) to deep learning approaches (Qiu et al., 2025b).

Table 1: Comparison of multimodal time series forecasting methods with text incorporation

| | Sub-categories | VoT (2025) | GPT4TS (2025) | TimeLLM (2024b) | TEST (2024) | CALF (2025a) | CMIN (2023) | Maformer (2024) | DualTime (2024) | Time-MMD (2024a) | TaTS (2026) | FNSPID (2024) | GPT4MTS (2024) | CiK (2025) |
|---|---|---|---|---|---|---|---|---|---|---|---|---|---|---|
| LLM Usage | Feature Extraction | ✓ | ✓ | ✓ | ✓ | ✓ | ✓ | ✓ | ✓ | ✓ | ✓ | ✓ | ✓ | ✓ |
| | Reasoning | ✓ | ✗ | ✗ | ✗ | ✗ | ✗ | ✗ | ✗ | ✗ | ✗ | ✗ | ✗ | ✗ |
| Text Type | Endogenous | ✓ | ✓ | ✓ | ✓ | ✓ | ✗ | ✗ | ✗ | ✗ | ✗ | ✗ | ✗ | ✓ |
| | Exogenous | ✓ | ✗ | ✗ | ✗ | ✗ | ✓ | ✓ | ✓ | ✓ | ✓ | ✓ | ✓ | ✓ |

These methods can be broadly categorized into MLP-based (Zeng et al., 2023; Yue et al., 2025), RNN-based (Chung et al., 2014; Wen et al., 2017; Cirstea et al., 2019), CNN-based (Sen et al., 2019; Liu et al., 2022; Wang et al., 2023), and GNN-based (Jin et al., 2024a; Cheng et al., 2024) models. Among them, Transformer-based models have become particularly prominent, evolving from foundational attention designs (Vaswani et al., 2017; Zhou et al., 2021) to decomposition-enhanced architectures (Wu et al., 2021; Zhou et al., 2022) and patch-based tokenization (Cirstea et al., 2022; Nie et al., 2023). Beyond architecture design, recent work has advanced training paradigms through frequency-domain learning and distribution alignment (Wang et al., 2025b;c), and broadened applicability to challenging settings such as probabilistic forecasting (Wu et al., 2025), non-stationary and irregular series (Liu et al., 2026) and exogenous variable integration (Qiu et al., 2025e). Automated architecture search (Wu et al., 2024), foundation models (Shi et al., 2024; Liu et al., 2024d;c; 2025b; Wang et al., 2024; Li et al., 2025), and general-purpose frameworks (Qiu et al., 2025c; Wang et al., 2025d; Zhang et al., 2026) further aim to provide unified solutions across diverse scenarios. These models primarily focus on the time series modality.

**Multimodal Time Series Forecasting with Text Incorporation.** Leveraging LLMs for time series has attracted growing interest. Methods such as GPT4TS (Chang et al., 2025), TimeLLM (Jin et al., 2024b), TEST (Sun et al., 2024), LLM-Mixer (Kowsher et al., 2025), CC-Time (Chen et al., 2025a), and CALF (Liu et al., 2025a) focus on enabling LLMs to understand time series through various encoding, alignment, and cross-modal fusion approaches. AimTS (Chen et al., 2025b) and TimeVLM (Zhong et al., 2025) further extend such efforts to visual modalities. However, these approaches primarily rely on temporal information without extensive exogenous text utilization. In domains rich in multimodal data (e.g., finance, health, and geoscience), CMIN (Luo et al., 2023), Modality-aware Transformer (Emami-Gohari et al., 2024), DualTime (Zhang et al., 2024), and multi-modal association methods (Qiu et al., 2025a) have started integrating exogenous texts with time series, while MM-TSFLib (Liu et al., 2024a) extends such integration across multiple domains. Building on this, TaTS (Li et al., 2026) reveals that time-aligned text and series exhibit similar periodicity. FNSPID (Dong et al., 2024) and GPT4MTS (Jia et al., 2024) place greater emphasis on text processing by filtering and summarizing text to extract salient signals, while CiK (Williams et al., 2025) proposes metrics for evaluating textual utility. More recently, ChatTime (Wang et al., 2025a) and Aurora (Wu et al., 2026) explore unified generative multimodal forecasting frameworks. As summarized in Table 1, existing methods either use LLMs only for feature extraction or handle only one text type (endogenous or exogenous). In contrast, our VoT jointly leverages LLMs for both feature extraction and reasoning while supporting both text types, enabling more comprehensive multimodal forecasting.

## 3 METHODOLOGY

### 3.1 PROBLEM FORMULATION

Consider a time series $\mathbf{X} = (\mathbf{x}_1, \mathbf{x}_2, ..., \mathbf{x}_L) \in \mathbb{R}^{L \times N}$, where $L$ denotes the length of the look-back window and $N$ represents the number of variables. Each time point $\mathbf{x}_t \in \mathbb{R}^N$ is associated with textual information $\mathbf{T}_t$. As mentioned in the introduction, we divide textual information $\mathbf{T}_t$ into two categories based on their source and characteristics.

**Exogenous text $\mathbf{T}^{\text{ex}}$** (Liu et al., 2024a) originates from external sources such as news articles, policy announcements, or social media posts, describing events, contexts, and factors that influence the time series beyond the system itself. **Endogenous text $\mathbf{T}^{\text{en}}$** (Kowsher et al., 2025) consists of structured textual descriptions derived directly from time series statistics.

The objective of multimodal time series forecasting is to predict the future $H$ time steps $\mathbf{Y} = (\mathbf{x}_{L+1}, \mathbf{x}_{L+2}, ..., \mathbf{x}_{L+H}) \in \mathbb{R}^{H \times N}$ by leveraging both historical observations and their associated

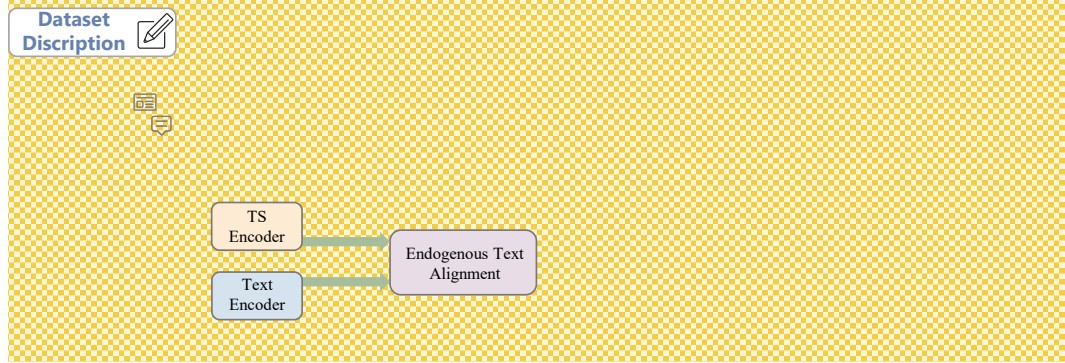

Figure 2: The architecture of VoT. The event-driven branch processes exogenous text through a three-step generative pipeline with the Historical In-Context Learning (HIC). The numerical branch aligns endogenous text with time series via the Endogenous Text Alignment (ETA). The Adaptive Frequency Fusion (AFF) combines both predictions across frequency bands with adaptive weights.

textual information:

$$\hat{\mathbf{Y}} = \mathcal{F}(\mathbf{X}, \mathbf{T}^{\text{ex}}, \mathbf{T}^{\text{en}}; \theta) \tag{1}$$

where $\hat{\mathbf{Y}} \in \mathbb{R}^{H \times N}$ represents the predicted values, $\mathcal{F}$ denotes the multimodal forecasting model, and $\theta$ encompasses all learnable parameters.

## 3.2 OVERVIEW

To maximize text utilization and enable mutual complementarity between textual and temporal information, we propose a dual-branch architecture that comprehensively leverages both exogenous and endogenous textual information. As illustrated in Figure 2, our method consists of two complementary branches that supports Event-driven Reasoning and Multi-level Alignment. The event-driven prediction branch primarily focuses on extracting predictive information from the exogenous text and the numerical prediction branch aligns time series with the generated endogenous text. For Event-driven Reasoning, the event-driven prediction branch extracts contextual information from exogenous text through a three-step generative pipeline. The pipeline is powered by the Historical In-Context Learning (HIC). Multi-level Alignment is designed to fully integrate the predictive capabilities of two modalities. At the representation level, the numerical prediction branch employs an Endogenous Text Alignment (ETA) to extract textual features aligned with time series patterns. At the prediction level, we introduce Adaptive Frequency Fusion (AFF). It fuses the frequency components of event-driven prediction and numerical prediction. Rather than assuming fixed influence of exogenous text on time series, AFF learns optimal fusion strategies directly from the dataset through Fourier decomposition and learnable frequency-specific weights.

## 3.3 EVENT-DRIVEN REASONING

To strengthen the reasoning capabilities of LLMs for time series forecasting, we propose a three-step generative pipeline including template generation, summarization, and reasoning. It converts exogenous text into event-driven numerical predictions enriched with contextual information. Additionally, we propose the Historical In-Context Learning (HIC). It keeps historical reasoning samples during training and retrieves similar examples as error-informed guidance during inference, as shown in Figure 3. The retrieved examples are corrected to provide guidance that helps avoid similar errors. By integrating these examples, LLMs are enhanced to generate accurate and error-informed predictions.

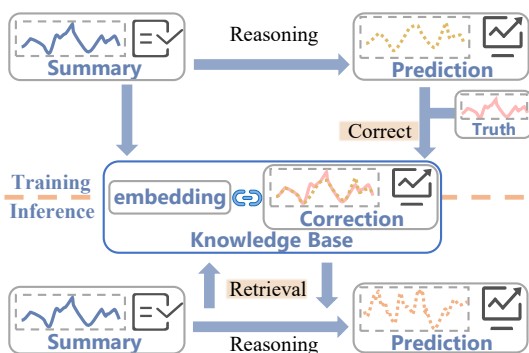

Figure 3: The processing procedure of the Historical In-Context Learning.

### 3.3.1 GENERATIVE PIPELINE

To process exogenous text $\mathbf{T}^{\text{ex}}$ into numerical predictions, we implement a three-step generative pipeline. First, LLM generates a dataset-specific template $\mathcal{D}$ based on the dataset description and exogenous text-time series samples, which serves as a structured dictionary containing key-value mappings that guide the extraction of predictive information. Second, we use this template $\mathcal{D}$ to generate summaries $\mathcal{S}_i$ from raw exogenous text $\mathbf{T}_i^{\text{ex}}$ and time series $\mathbf{X}_i$, filtering redundant information while preserving the information influential to time series forecasting. Finally, we employ the Reasoner, an LLM in the form of a reasoning model, specialized for logical deduction and numerical forecasting rather than general natural language generation. It process these summaries and generate predictions:

$$\hat{\mathbf{Y}}_i^{\text{event}}, \mathcal{R}_i = \text{Reasoner}(\mathcal{P}_{\text{reason}}, \mathcal{S}_i, \mathbf{X}_i) \tag{2}$$

The reasoning model Reasoner generates both the numerical prediction $\hat{\mathbf{Y}}_i^{\text{event}} \in \mathbb{R}^{H \times N}$ and the explanatory reasoning process $\mathcal{R}_i$. $\mathcal{P}_{\text{reason}}$ is a carefully designed prompt that guides the LLM to perform structured reasoning from summaries $\mathcal{S}_i$ and time series $\mathbf{X}_i$ to numerical predictions. An example in Appendix E showing $\mathcal{P}_{\text{reason}}$, $\mathcal{S}_i$ and $\mathcal{R}_i$ can be quite helpful for understanding the roles of these items. However, this basic pipeline operates in an unsupervised manner, which may introduce suboptimal guidance for numerical prediction patterns, potentially amplifying prediction errors. Accordingly, we design the Historical In-Context Learning (HIC) to solve the problem.

### 3.3.2 HISTORICAL IN-CONTEXT LEARNING

To make the prediction more accurate, we introduce Historical In-Context Learning (HIC), which synergizes historical reasoning information with In-Context Learning (ICL), as shown in Figure 3. During training, it corrects the reasoning process with ground-truth and keeps the corrected samples. The optimized reasoning process of the corrected samples keeps information about why errors exist and correct reasoning strategies. During inference, it retrieves the most similar historical example and integrates it into the reasoning prompt to obtain error-informed guidance and accurate forecast.

**Knowledge Base Construction.** During training, HIC builds a Knowledge Base $\mathcal{K}$ by learning from prediction errors. As mentioned, the Reasoner generates initial predictions $\hat{\mathbf{Y}}_i^{\text{event}}$ and reasoning processes $\mathcal{R}_i$. After that, the Reasoner creates corrected reasoning $\mathcal{C}_i$ using ground truth $\mathbf{Y}_i$:

$$\mathcal{C}_i = \text{Reasoner}(\mathcal{P}_{\text{correct}}, \hat{\mathbf{Y}}_i^{\text{event}}, \mathcal{R}_i, \mathbf{Y}_i, \mathbf{X}_i) \tag{3}$$

where $\mathcal{P}_{\text{correct}}$ is the prompt that guides the Reasoner to identify and understand what caused the error between the initial predictions $\hat{\mathbf{Y}}_i^{\text{event}}$ and the ground-truth $\mathbf{Y}_i$. Correction $\mathcal{C}_i$ explains how to accurately derive the actual values given the context, and crucially, it contains analysis of the previous prediction errors, which helps Reasoner better understand and avoid similar errors in the future. The Knowledge Base $\mathcal{K}$ stores pairs of summary embeddings of summaries $\{\mathcal{S}_i\}_{i=1}^M$ and their corresponding correction $\{\mathcal{C}_i\}_{i=1}^M$:

$$\mathcal{K} = \{(\text{Embed}(\mathcal{S}_i), \mathcal{C}_i)\}_{i=1}^M \tag{4}$$

where $M$ denotes the number of training samples, Embed is an embedding model.

**Retrieval-Guided Prediction.** During inference, HIC retrieves the most similar historical example from $\mathcal{K}$ to obtain error-informed guidance for forecasting. Given a data pair $(\mathbf{X}_j, \mathbf{T}_j^{\text{ex}})$, first LLM generates the summary $\mathcal{S}_j$. Then using the embedding of the summary $\mathcal{S}_j$, HIC retrieves the most similar example in the Knowledge Base:

$$\tilde{i} = \arg \max_{(\mathcal{S}_i, \mathcal{C}_i) \in \mathcal{K}} \text{simi}(\text{Embed}(\mathcal{S}_j), \text{Embed}(\mathcal{S}_i)) \tag{5}$$

where simi is the similarity metric. The retrieved corresponding correction $\mathcal{C}_{\tilde{i}}$ serves as an in-context example, improving the reasoning accuracy with error-informed guidance:

$$\hat{\mathbf{Y}}_j^{\text{event}} = \text{Reasoner}(\mathcal{P}_{\text{ICL}}, \mathcal{C}_{\tilde{i}}, \mathcal{S}_j, \mathbf{X}_j) \tag{6}$$

where $\mathcal{P}_{\text{ICL}}$ is the prompt that combines the retrieved correction $\mathcal{C}_{\tilde{i}}$ as guidance for current inference.

By retrieving historically corrected reasoning patterns, HIC provides error-informed guidance for LLMs. The error-informed learning helps the model understand how exogenous text impacts time

series, thereby improving predictions when similar event-driven fluctuations occur. Moreover, HIC achieves this without requiring expensive fine-tuning, making it efficient and scalable across different forecasting domains.

Event-driven Reasoning generates numerical predictions by leveraging semantic information in exogenous text and capturing event-driven dynamics. Through correcting the reasoning process, constructing a knowledge base, and implementing a retrieve-and-guide mechanism, our approach enhances the reasoning ability of LLM while maximizing text utilization.

### 3.4 MULTI-LEVEL ALIGNMENT

While the event-driven prediction branch excels at capturing external influences through exogenous text, not all temporal variations are reflected in or captured by exogenous text. Some data follow intrinsic patterns that are better captured through numerical analysis. To fully leverage the advantages of both modalities, we design a Multi-level Alignment method. It performs representation-level alignment with endogenous text via the Endogenous Text Alignment (ETA) of the numerical branch. In the prediction level, it employs Adaptive Frequency Fusion (AFF) to align exogenous text prediction with numerical prediction and integrate results from both branches . The outputs obtained after deep alignment obtain the complementary advantages of both modalities.

#### 3.4.1 REPRESENTATION-LEVEL ALIGNMENT

To achieve representation-level alignment, the numerical prediction branch employs an Endogenous Text Alignment (ETA), as shown in Figure 4. This module establishes deep semantic alignment between temporal patterns and their textual representations using decomposed pattern extraction and decomposed contrastive learning. Considering that trend and seasonality are intrinsic properties of time series, ETA similarly extracts corresponding textual representations to achieve fine-grained alignment between endogenous text and temporal patterns.

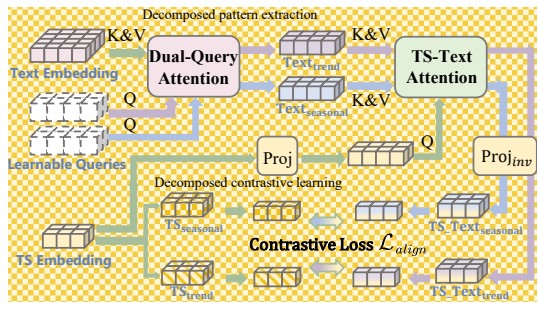

Figure 4: The processing procedure of the Endogenous Text Alignment (ETA).

**Time Series and Text Encoding.** We encode the input time series $\mathbf{X} \in \mathbb{R}^{L \times N}$ using an encoder to obtain temporal representations $\mathbf{H}^{\text{ts}} \in \mathbb{R}^{N \times d^{\text{ts}}}$. Simultaneously, we generate endogenous text $\mathbf{T}^{\text{en}}$ by converting statistical descriptors such as mean and frequency into structured textual descriptions, which are then encoded using an LLM to obtain text embeddings $\mathbf{H}^{\text{text}} \in \mathbb{R}^{L^{\text{text}} \times d^{\text{text}}}$, where $L$ and $L^{\text{text}}$ denote the sequence length and text token length respectively, $N$ is the number of variables, and $d^{\text{ts}}$, $d^{\text{text}}$ are the embedding dimensions for time series and text respectively.

**Decomposed Pattern Extraction.** Our decomposed pattern extraction first employs the dual-query attention to filter semantic components from text. First, we use learnable queries $\mathbf{Q}^{\text{tr}}, \mathbf{Q}^{\text{se}} \in \mathbb{R}^{N \times d^{\text{text}}}$ to extract trend and seasonal information $\mathbf{E}^{\text{tr}}, \mathbf{E}^{\text{se}} \in \mathbb{R}^{N \times d^{\text{text}}}$ related to time series from textual representations:

$$\mathbf{E}^{\text{tr}} = \text{Attention}(\mathbf{Q}^{\text{tr}}, \mathbf{H}^{\text{text}}, \mathbf{H}^{\text{text}}), \quad \mathbf{E}^{\text{se}} = \text{Attention}(\mathbf{Q}^{\text{se}}, \mathbf{H}^{\text{text}}, \mathbf{H}^{\text{text}}) \quad (7)$$

Then, the ETA performs ts-text attention to align temporal representations with extracted textual components, obtaining further aligned representations $\hat{\mathbf{Z}}^{tr}$ and $\hat{\mathbf{Z}}^{se}$ (cross-modal aligned features):

$$\mathbf{Z}^* = \text{Cross-Attention}(\text{Proj}(\mathbf{H}^{\text{ts}}), \mathbf{E}^*, \mathbf{E}^*), \quad * \in \{\text{tr}, \text{se}\} \quad (8)$$

where $\text{Proj}(\cdot) : \mathbb{R}^{d^{\text{ts}}} \to \mathbb{R}^{d^{\text{text}}}$ projects time series embeddings to the text embedding space, and $\mathbf{Z}^{\text{tr}}, \mathbf{Z}^{\text{se}} \in \mathbb{R}^{N \times d^{\text{text}}}$ are the aligned trend and seasonal representations that combine information from both modalities. To map these representations to the time series space for fusion, we apply:

$$\tilde{\mathbf{Z}}^* = \text{Proj}_{\text{inv}}(\mathbf{Z}^*), \quad * \in \{\text{tr}, \text{se}\} \quad (9)$$

where $\text{Proj}_{\text{inv}}(\cdot) : \mathbb{R}^{d^{\text{text}}} \to \mathbb{R}^{d^{\text{ts}}}$, yielding $\tilde{\mathbf{Z}}^{\text{tr}}, \tilde{\mathbf{Z}}^{\text{se}} \in \mathbb{R}^{N \times d^{\text{ts}}}$.

**Deep Semantic Alignment via Decomposed Contrastive Learning.** To achieve deep semantic alignment between temporal patterns and textual representations, we employ contrastive learning at the sample level. We first decompose the temporal representations $\mathbf{H}_i^{\text{ts}}$ into trend and seasonal components to obtain $\mathbf{H}_i^{\text{tr}}$ and $\mathbf{H}_i^{\text{se}}$. Then, we get the mean representation $\bar{\mathbf{H}}_i^{\text{tr}}, \bar{\mathbf{H}}_i^{\text{se}} \in \mathbb{R}^{d^{\text{ts}}}$ and $\bar{\mathbf{Z}}_i^{\text{tr}}, \bar{\mathbf{Z}}_i^{\text{se}} \in \mathbb{R}^{d^{\text{ts}}}$. We compute the contrastive loss for each component pair.

$$\mathcal{L}_{\text{align}} = \frac{1}{2} \sum \left( -\log \frac{\exp(\text{sim}(\bar{\mathbf{H}}_i^*, \bar{\mathbf{Z}}_i^*))}{\sum_{j=1}^{B} \exp(\text{sim}(\bar{\mathbf{H}}_i^*, \bar{\mathbf{Z}}_j^*))} - \log \frac{\exp(\text{sim}(\bar{\mathbf{Z}}_i^*, \bar{\mathbf{H}}_i^*))}{\sum_{j=1}^{B} \exp(\text{sim}(\bar{\mathbf{Z}}_i^*, \bar{\mathbf{H}}_j^*))} \right), \quad * \in \{\text{tr}, \text{se}\} \tag{10}$$

where $\text{sim}(\cdot, \cdot)$ denotes cosine similarity and $B$ is the batch size. The total alignment loss combines both trend and seasonal components. This objective ensures that corresponding trend and seasonal components from both modalities are aligned in a shared representation space.

After completing the modal alignment, the numerical prediction output $\mathbf{Y}_{num}$ is generated by fusing temporal and text representations. Specifically, the calculation formula of $\mathbf{Y}_{num}$ is defined as $\mathbf{Y}_{num} = \frac{1}{2}\mathbf{H}^{\text{ts}} + \frac{1}{2}(\tilde{\mathbf{Z}}^{\text{tr}} + \tilde{\mathbf{Z}}^{\text{se}})$, where $\mathbf{H}^{\text{ts}}$ represents the original temporal representation, and $(\tilde{\mathbf{Z}}^{\text{tr}}, \tilde{\mathbf{Z}}^{\text{se}})$ denotes the text-derived representations corresponding to trend and seasonal components respectively. The fusion process adopts equal weight allocation to balance the contributions of temporal and textual information.

### 3.4.2 PREDICTION-LEVEL ALIGNMENT

Event-driven predictions excel at capturing patterns influenced by external factors, while not all temporal variations are reflected in or captured by textual information. These complementary strengths can potentially be integrated through frequency-based approaches. Therefore, at the prediction level, we introduce Adaptive Frequency Fusion (AFF) to dynamically adjust the importance of different frequency components, thereby leveraging the complementary strengths of textual and numerical information across frequency bands.

**Adaptive Frequency Fusion.** We decompose both branch predictions into frequency components:

$$\mathcal{F}^{\text{num}} = \text{FFT}(\hat{\mathbf{Y}}^{\text{num}}), \quad \mathcal{F}^{\text{event}} = \text{FFT}(\hat{\mathbf{Y}}^{\text{event}}) \tag{11}$$

The spectrum is partitioned into three bands based on frequency. For the detailed adaptive partitioning strategy and full implementation details, we refer the reader to Appendix A.4. We extract band-specific components with mask $\mathbf{M}$:

$$\mathcal{F}_*^b = \mathcal{F}_* \odot \mathbf{M}^b, \quad * \in \{\text{num}, \text{event}\}, \quad b \in \{\text{low}, \text{mid}, \text{high}\} \tag{12}$$

Instead of using fixed fusion ratios, we introduce learnable weights $\mathbf{w} = w_*^b, * \in \{\text{num}, \text{event}\}, b \in \{\text{low}, \text{mid}, \text{high}\}$ that adapt to data characteristics:

$$\mathcal{F}_{\text{fused}} = \sum_* \sum_b w_*^b \mathcal{F}_*^b, * \in \{\text{num}, \text{event}\}, b \in \{\text{low}, \text{mid}, \text{high}\}, \quad \hat{\mathbf{Y}}_{\text{final}} = \text{iFFT}(\mathcal{F}_{\text{fused}}) \tag{13}$$

**Training Objective.** Our model is trained with a composite loss function:

$$\mathcal{L}_{\text{total}} = \mathcal{L}_{\text{ts}} + \mathcal{L}_{\text{align}} + \mathcal{L}_{\text{final}} \tag{14}$$

where $\mathcal{L}_{\text{ts}} = \text{MSE}(\hat{\mathbf{Y}}_{\text{ts}}, \mathbf{Y})$ maintains base temporal prediction capability and $\hat{\mathbf{Y}}_{\text{ts}}$ is from the numerical branch without ETA. $\mathcal{L}_{\text{align}}$ enforces cross-modal alignment via contrastive learning, and $\mathcal{L}_{\text{final}} = \text{MSE}(\hat{\mathbf{Y}}_{\text{final}}, \mathbf{Y})$ optimizes the fused prediction accuracy.

## 4 EXPERIMENTS

### 4.1 EXPERIMENTAL SETUP

**Datasets.** We evaluate VoT on 10 real-world multimodal time series datasets. 9 datasets are sourced from the MM-TSFLib benchmark (Liu et al., 2024a), covering diverse domains including Agriculture, Climate, Economy, Energy, Public Health (United States), Environment, Traffic, and Security.

Table 2: Forecasting results of time series-only and text-enhanced methods, and our method VoT. The best results are highlighted in **bold**, and the second-best results are underlined.

| Category | | | Time series-only | | | | | | Text-enhanced | | | | | |
|---|---|---|---|---|---|---|---|---|---|---|---|---|---|---|
| Models | VoT | | PatchTST (2023) | | iTransformer (2024b) | | RaFT (2025) | | PatchTST* | | iTransformer* | | RaFT* | |
| Metric | MSE | MAE | MSE | MAE | MSE | MAE | MSE | MAE | MSE | MAE | MSE | MAE | MSE | MAE |
| Agriculture | **0.209** | **0.302** | 0.228 | 0.303 | 0.220 | 0.308 | 0.226 | 0.322 | 0.232 | 0.316 | 0.229 | 0.310 | 0.246 | 0.333 |
| Climate | **1.078** | **0.840** | 1.184 | 0.888 | 1.135 | 0.865 | 1.289 | 0.926 | 1.178 | 0.887 | 1.117 | 0.858 | 1.342 | 0.944 |
| Economy | **0.201** | **0.353** | 0.210 | 0.363 | 0.222 | 0.378 | 0.265 | 0.411 | 0.219 | 0.370 | 0.213 | 0.367 | 0.275 | 0.420 |
| Energy | **0.222** | **0.343** | 0.250 | 0.363 | 0.269 | 0.382 | 0.254 | 0.367 | 0.253 | 0.365 | 0.265 | 0.383 | 0.246 | 0.360 |
| Environment | **0.268** | **0.380** | 0.317 | 0.395 | 0.276 | 0.386 | 0.339 | 0.423 | 0.318 | 0.397 | 0.278 | 0.390 | 0.337 | 0.422 |
| Health | **1.205** | **0.714** | 1.432 | 0.804 | 1.519 | 0.833 | 1.833 | 0.975 | 1.360 | 0.768 | 1.713 | 0.915 | 1.788 | 0.963 |
| Security | **70.117** | **3.937** | 72.027 | 4.062 | 75.042 | 4.217 | 77.204 | 4.473 | 72.721 | 4.177 | 74.032 | 4.154 | 76.587 | 4.448 |
| Social Good | **0.804** | **0.389** | 0.944 | 0.475 | 0.961 | 0.463 | 0.968 | 0.484 | 0.909 | 0.427 | 1.027 | 0.515 | 0.970 | 0.477 |
| Traffic | **0.169** | **0.232** | 0.176 | 0.234 | 0.184 | 0.238 | 0.288 | 0.382 | 0.174 | 0.239 | 0.184 | 0.237 | 0.300 | 0.394 |
| Weather | **0.968** | **0.706** | 1.145 | 0.751 | 1.231 | 0.803 | 1.099 | 0.746 | 1.036 | 0.707 | 1.004 | 0.709 | 1.096 | 0.745 |
| 1st counts | **20** | | 0 | | 0 | | 0 | | 0 | | 0 | | 0 | |

Table 3: Forecasting results of multimodal methods and VoT. The best results are highlighted in **bold**, and the second-best results are underlined.

| Models | VoT | | GPT4TS (2025) | | GPT4MTS (2024) | | TaTS (2026) | | Time-VLM (2025) | | CALF (2025a) | |
|---|---|---|---|---|---|---|---|---|---|---|---|---|
| Metric | MSE | MAE | MSE | MAE | MSE | MAE | MSE | MAE | MSE | MAE | MSE | MAE |
| Agriculture | **0.209** | 0.302 | 0.220 | **0.294** | 0.225 | 0.298 | 0.215 | 0.301 | 0.238 | 0.303 | 0.250 | 0.315 |
| Climate | **1.078** | **0.840** | 1.184 | 0.891 | 1.182 | 0.889 | 1.180 | 0.887 | 1.195 | 0.899 | 1.286 | 0.922 |
| Economy | **0.201** | **0.353** | 0.217 | 0.371 | 0.208 | 0.363 | 0.215 | 0.368 | 0.229 | 0.384 | 0.207 | 0.357 |
| Energy | **0.222** | **0.343** | 0.260 | 0.376 | 0.262 | 0.380 | 0.255 | 0.368 | 0.260 | 0.374 | 0.244 | 0.365 |
| Environment | **0.268** | **0.380** | 0.322 | 0.393 | 0.323 | 0.400 | 0.319 | 0.396 | 0.325 | 0.398 | 0.325 | 0.387 |
| Health | **1.205** | **0.714** | 1.341 | 0.777 | 1.464 | 0.799 | 1.356 | 0.767 | 1.490 | 0.835 | 1.491 | 0.775 |
| Security | **70.117** | **3.937** | 71.165 | 4.047 | 71.487 | 4.068 | 72.406 | 4.097 | 73.731 | 4.182 | 76.376 | 4.300 |
| Social Good | **0.804** | **0.389** | 0.917 | 0.476 | 0.920 | 0.450 | 0.918 | 0.428 | 0.869 | 0.444 | 0.906 | 0.401 |
| Traffic | **0.169** | **0.232** | 0.206 | 0.266 | 0.203 | 0.261 | 0.179 | 0.238 | 0.217 | 0.320 | 0.222 | 0.293 |
| Weather | **0.968** | **0.706** | 1.048 | 0.708 | 0.986 | 0.711 | 1.037 | **0.706** | 1.061 | 0.717 | 1.098 | 0.714 |
| 1st counts | **19** | | 1 | | 0 | | 1 | | 0 | | 0 | |

Notably, the original Economy dataset in MM-TSFLib is arranged in reverse temporal order. We reorder this dataset temporally to align with the natural progression of events and maintain temporal consistency. Additionally, we introduce a new Weather dataset containing multimodal meteorological observations. We follow the experimental settings from MM-TSFlib (Liu et al., 2024a) for dataset preprocessing and forecasting configurations. Detailed dataset descriptions are provided in the Appendix A.1. Notably, we do not apply the "Drop Last" trick to ensure a fair comparison following the settings (Qiu et al., 2024).

**Baselines.** We compare VoT against eleven representative baselines: three time series-only methods including iTransformer (Liu et al., 2024b), PatchTST (Nie et al., 2023), and RAFT (Han et al., 2025); three text-enhanced variants iTransformer*, PatchTST*, and RAFT* that incorporate textual information through the Time-MMD framework (Liu et al., 2024a); and five multimodal forecasting methods including GPT4TS (Chang et al., 2025), GPT4MTS (Jia et al., 2024), TaTS (Li et al., 2026), Time-VLM (Zhong et al., 2025) and CALF (Liu et al., 2025a). All baselines are implemented using their official code repositories when available, with consistent experimental settings and hyperparameter tuning on the validation set.

**Metrics.** We adopt two standard metrics in time series forecasting: Mean Squared Error (MSE) and Mean Absolute Error (MAE). MSE emphasizes larger errors and is more sensitive to outliers, while MAE provides a more interpretable measure of average prediction error. We report averaged results across all prediction horizons for comprehensive evaluation in section 4.2, with detailed results for individual horizons provided in Appendix G.

We conduct a comparison of different reasoning-focused LLMs (Reasoner variants) in Appendix F, and all of them show consistent effectiveness in our forecasting framework.

## 4.2 MAIN RESULTS

Tables 2 and 3 present the outcomes of our comprehensive evaluation across 10 real-world multimodal time series datasets. Our method VoT achieves the best performance on nearly all datasets, ranking first in all 20 metrics against time series-only and text-enhanced baselines, and 19 out of 20 metrics against multimodal methods. This dominant performance across diverse domains demonstrates the effectiveness of our Event-driven Reasoning and Multi-level Alignment approach in lever-

| | | TS-only | w/o ETA | w/o HIC | w/o Event | VoT |
|---|---|---|---|---|---|---|
| Energy | MSE | 0.250 | 0.241 | 0.238 | 0.243 | **0.222** |
| | MAE | 0.363 | 0.355 | 0.363 | 0.360 | **0.343** |
| Social Good | MSE | 0.944 | 0.840 | 0.845 | 0.876 | **0.804** |
| | MAE | 0.475 | 0.424 | 0.410 | 0.436 | **0.389** |

Table 4: Ablation study results on Energy and Social Good datasets

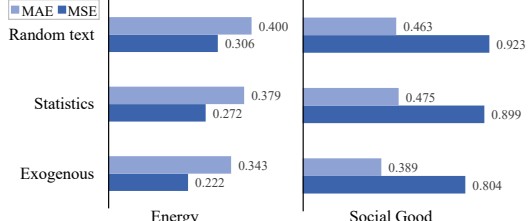

Figure 5: Ablation study on different text sources for event-driven reasoning.

aging both exogenous and endogenous textual information. Notably, our method demonstrates superior performance across diverse domains, particularly excelling on datasets that are susceptible to event influence and rich in exogenous textual information, such as Energy and Social Good.

## 4.3 MODEL ANALYSIS

### 4.3.1 ABLATION STUDIES

To validate the effectiveness of each component in our model, we conduct ablation studies on the Energy and Social Good datasets. Table 4 presents the ablation results. The TS-only baseline achieves 0.250/0.944 MSE on Energy/Social Good, demonstrating significant performance gaps compared to our full method (0.222/0.804). Removing ETA (w/o ETA) degrades performance to 0.241/0.840 MSE, confirming that structured textual descriptions enhance pattern recognition. Without HIC (w/o HIC), performance drops to 0.238/0.845 MSE, with more significant degradation on Social Good, validating that the retrieve-and-guide mechanism is crucial for event-driven forecasting. Without it, the LLM cannot be effectively guided in its reasoning process. Notably, removing HIC results in worse performance than removing the entire event-driven branch (w/o Event), suggesting that unguided LLM reasoning can be more detrimental than no reasoning at all. When equipped with proper guidance, the event-driven branch becomes highly effective, as evidenced by the significant performance gap between our full model (0.222/0.804) and the w/o Event baseline (0.232/0.841).

### 4.3.2 EXOGENOUS TEXT IMPACT ANALYSIS

To verify whether the event-driven branch of VoT truly captures semantic information from text to enhance time series forecasting, we conduct a text replacement experiment. Specifically, we compare our approach using exogenous text with two alternatives: randomly generated text and statistics-based text derived from the time series, where the latter refers to the endogenous text in this paper. As shown in Figure 5, results clearly demonstrate the superiority of exogenous text across both datasets. On the Energy dataset, exogenous text achieves 27.5% lower MSE and 14.3% lower MAE compared to random text. Similar improvements are observed on the Social Good dataset with 12.8% MSE reduction and 16.0% MAE reduction. The statistics show intermediate performance, suggesting that while time series-derived features provide some value, they cannot match the rich contextual information from real-world exogenous sources. This confirms that it is of great significance for us to propose an event-driven prediction branch and incorporate exogenous texts into the multimodal time series forecasting methods.

### 4.3.3 ENDOGENOUS TEXT ALIGNMENT ANALYSIS

The reason for adopting ETA is that decomposition is a simple and widely-used technique in time series analysis. By decomposing the time series into trend and seasonal components, we can enrich the semantic representation of the time series to a certain degree. This enrichment allows for better alignment with textual information, facilitating more effective multi-modal interaction. To further validate the effectiveness of this approach, we conducted supplementary ablation experiments on the numerical branch. The "w/o decomposition" condition indicates the absence of trend and seasonal decomposition, with only standard TS-Text contrastive learning being performed. The "w/o TS-Text

CL" condition means that only time series decomposition is applied, without TS-Text contrastive learning. The experimental results, as presented in Table 5, demonstrate that our proposed method, ETA, achieves superior performance compared to alternative designs.

Table 5: Ablation Study on Components of the Event-based Temporal Alignment (ETA)

| | Climate | | Energy | | SocialGood | |
|---|---|---|---|---|---|---|
| **Variant** | MSE | MAE | MSE | MAE | MSE | MAE |
| w/o Decomposition | 1.120 | 0.859 | 0.254 | 0.368 | 0.892 | 0.440 |
| w/o TS-Text CL | 1.184 | 0.888 | 0.250 | 0.363 | 0.944 | 0.475 |
| **ETA** | **1.092** | **0.848** | **0.232** | **0.350** | **0.841** | **0.410** |

### 4.3.4 ADAPTIVE FREQUENCY FUSION ANALYSIS

To further validate the necessity of the AFF, we conduct frequency-domain analysis on the Social Good dataset, as shown in Figure 6. We apply different frequency filters to analyze the frequency characteristics of each branch. In Figure 6 (b), the low-pass filtered signals (0-10% frequencies), the event-driven branch aligns more closely with the ground truth than the TS-only branch, demonstrating its strength in capturing trend patterns and event-induced shifts; In Figure 6 (c), the high-pass filtered signals (70-100% frequencies), the TS-only method better matches the ground truth, effectively capturing short-term fluctuations and periodic patterns; In Figure 6 (d), the band-pass filtered signals (10-70% frequencies) reveal increased similarity among all methods, indicating dissimilarity is much more prevalent in high and low frequencies. The choice of these frequency thresholds and the corresponding sensitivity analysis to the thresholds are provided in Appendix A.4.These observations empirically confirm that different prediction branches excel in distinct frequency bands, highlighting the need for frequency fusion methods. Since different datasets exhibit varying degrees of correlation with events, resulting in different frequency component distributions, adaptive fusion is necessary. This validates the necessity and effectiveness of our AFF over fixed combination strategies.

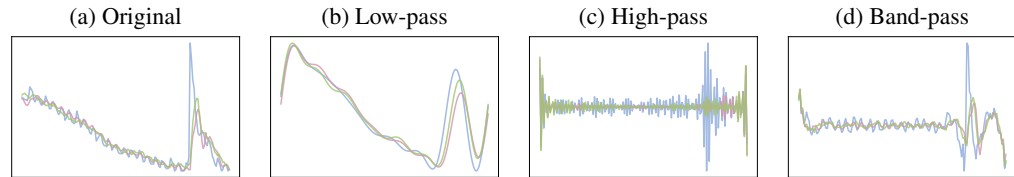

Figure 6: Frequency domain analysis of time series predictions (Social Good). (a) Original signals without frequency decomposition. (b)-(d) Frequency-filtered components: (b) Low-pass filtered signals, (c) High-pass filtered signals, and (d) Band-pass filtered signals. Ground Truth (blue), Time series-only prediction (pink), and event-driven prediction (green)
.

## 5 CONCLUSION

In this paper, we presented VoT, a multimodal time series forecasting method that unlocks the value of textual information through Event-driven Reasoning and Multi-level Alignment. Our approach leverages both exogenous and endogenous text via a dual-branch architecture, where Historical In-Context Learning enables LLMs to learn from historical prediction errors, Endogenous Text Alignment bridges the semantic gap between modalities, and Adaptive Frequency Fusion dynamically combines their complementary strengths. Extensive experiments across 10 real-world datasets demonstrate that VoT achieves state-of-the-art performance, particularly excelling on event-influenced domains. Our work establishes that textual information provides irreplaceable contextual guidance for time series forecasting, and that effectively integrating these textual contexts with numerical patterns is crucial for advancing beyond the limitations of single-modality approaches.

## ACKNOWLEDGEMENTS

This work was partially supported by the National Natural Science Foundation of China (62406112, 62372179). Yang Shu is the corresponding author of the work.

## ETHICS STATEMENT

Our work exclusively uses publicly available benchmark datasets that contain no personally identifiable information. The proposed anomaly detection framework is designed for beneficial applications in system reliability and safety monitoring. No human subjects were involved in this research.

## REPRODUCIBILITY STATEMENT

The performance of VoT and datasets used in our work are real, and all experimental results can be reproduced. We have released our model code and checkpoints at https://github.com/decisionintelligence/VoT.

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

# A  EXPERIMENTAL DETAILS

## A.1  DATASETS

Our experiments utilize 10 multimodal time series datasets from two sources: **MM-TSFLib Datasets (Liu et al., 2024a):** We adopt 9 datasets from the MM-TSFLib benchmark, maintaining their original preprocessing and alignment procedures. These datasets span:

- **Agriculture**: USDA broiler market prices with weekly market reports

- **Climate**: NOAA drought indices with monthly climate reports

- **Economy**: US international trade balance (chronologically reordered for our experiments)

- **Energy**: Weekly US gasoline prices from EIA

- **Health**: CDC influenza-like illness statistics

- **Environment**: New York EPA air quality measurements

- **Traffic**: FHWA vehicle miles traveled statistics

- **Security**: FEMA disaster declarations

**Weather Dataset:** We introduce a new multimodal dataset containing hourly observations of weather. Each data point includes four numeric variables (MINTEMP, MAXTEMP, HUMIDITY, MAXHUMIDITY(OT)) aligned with natural language weather descriptions. The OT variable represents the maximum humidity in local regions.

We follow the experimental settings from MM-TSFLib (Liu et al., 2024a), with configurations varying based on data reporting frequency:

For Environment and Weather datasets, we use a lookback window of $L = 96$ time steps, with forecasting horizons $H \in \{48, 96, 192, 336\}$ and a label window size of 48 for the decoder.

For Health and Energy datasets, the lookback window is set to $L = 36$, with forecasting horizons $H \in \{12, 24, 36, 48\}$ and a label window size of 18.

For Agriculture, Climate, Economy, Security, Social Good, and Traffic datasets, we employ a lookback window of $L = 8$, forecasting horizons $H \in \{6, 8, 10, 12\}$, and a label window size of 4.

## A.2  DATASET SPLIT AND TIME LEAKAGE MITIGATION

Our dataset split is time-based. The first 70% of the data in temporal order serves as the training set, the 70%-80% segment serves as the validation set, and the latest 20% serves as the test set. There is no temporal overlap between the training, validation, and test sets. When constructing the knowledge base, we exclusively use relevant data from the training set. During the inference phase, that is, when the input is from the validation or test set, retrieving examples from the knowledge database only returns training-set data. All these data are temporally prior to the data in the validation and test sets, so no time leakage occurs.

## A.3  BASELINES

We evaluate our approach against nine representative baselines spanning from pure time series methods to multimodal forecasting approaches. The baselines are selected based on their state-of-the-art performance and represent diverse architectural designs from recent years. The specific code repositories for each model are shown in Table 6

For the text-enhanced variants (marked with *), we integrate textual information following the Time-MMD framework implementation, which concatenates text embeddings with time series representations before the prediction layer. All models are evaluated under identical experimental conditions with consistent data preprocessing and hyperparameter tuning protocols.

Table 6: Implementation details and code repositories for baseline methods

| Category | Method | Repository |
|---|---|---|
| Time Series Only | iTransformer | https://github.com/thuml/iTransformer |
| | PatchTST | https://github.com/yuqinie98/PatchTST |
| | RAFT | https://github.com/archon159/RAFT |
| Text-Enhanced (Time-MMD) | iTransformer* | Based on iTransformer with Time-MMD text integration |
| | PatchTST* | Based on PatchTST with Time-MMD text integration |
| | RAFT* | Based on RAFT with Time-MMD text integration |
| | Time-MMD | https://github.com/AdityaLab/MM-TSFlib |
| Multimodal Forecasting | GPT4TS | https://github.com/blacksnail789521/LLM4TS |
| | GPT4MTS | https://github.com/Flora-jia-jfr/GPT4MTS-Prompt-based-Large-Language-Model-for-Multimodal-timeseries-Forecasting |
| | TaTS | https://github.com/iDEA-iSAIL-Lab-UIUC/TaTS |
| | TimeVLM | https://github.com/CityMind-Lab/ICML25-TimeVLM |
| | CALF | https://github.com/Hank0626/CALF |

### A.4 DETAILED IMPLEMENTATION SETTINGS

**Software Environment.** Our implementation uses PyTorch 2.5.0+cu121 with Python 3.10. All experiments are conducted on Ubuntu 20.04 with CUDA 12.1.

**Event-Driven Branch Configuration.**

- **Template Generation and Summarization**: We employ Ollama-wrapped Meta-Llama-3-8B-Instruct for generating dataset-specific templates and extracting event summaries from exogenous text.
- **Knowledge Base Construction**: Historical patterns are embedded using Qwen2-Embedding-0.6B (768-dimensional vectors) for efficient similarity retrieval.
- **Reasoning and Optimization**: During inference, we utilize the DeepSeek API for generating predictions and performing reasoning correction. The most similar historical patterns is retrieved based on cosine similarity.

**Numerical Branch Configuration.**

- **Backbone Architecture**: PatchTST with 2-layer encoder (e_layers=2).
- **Endogenous Text Generation**: GPT-2 converts structured textual descriptions into representation.

**Adaptive Frequency Fusion Settings.**

- **Band Partition**: Initial partition uses Low (0-10%), Mid (10-70%), High (70-100%) frequency bands. The actual boundaries are adaptively adjusted based on the frequency spectrum characteristics of each dataset to ensure effective separation between Low, Mid, and High components.

  We conducted tests under different boundary conditions to evaluate the sensitivity of the model and presented the results in Figure 7 of the updated version. The figure displays a heat map of MSE or MAE values with respect to different low- and high-frequency boundaries. Through analysis of the results, we observe that with the change of low and high frequency bounds, the results do not change significantly when the boundary selection is reasonable, indicating low boundary parameter sensitivity.
- **Weight Initialization**: All six fusion weights $\mathbf{w} = [0.5, 0.5, 0.5, 0.5, 0.5, 0.5]$ for balanced initial contributions from both branches across all frequency bands.

**Multi-Stage Training Protocol.**

- **Stage 1 (Pre-training)**: Train PatchTST backbone for 10 epochs with MSE loss, learning rate 1e-4, cosine annealing scheduler.
- **Stage 2 (Alignment)**: Train alignment module for 10 epochs with contrastive loss, learning rate 1e-3, encoder weights frozen.

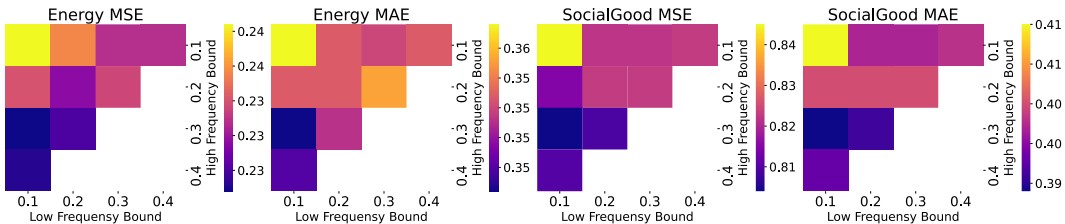

Figure 7: Heat Map of MSE/MAE Values Based on Low-Frequency and High-Frequency Percentage Cutoffs. The x-axis represents the percentage cutoffs for the low-frequency range, varying from 10% to 40% of the low-frequency spectrum. The y-axis represents the percentage cutoffs for the high-frequency range, ranging from 10% to 40% of the high-frequency spectrum. The color intensity in the map indicates the magnitude of the Mean Squared Error (MSE) or Mean Absolute Error (MAE)

- **Stage 3** (**Joint Optimization**): Fine-tune all components for 30 epochs, learning rate selected from {5e-4, 1e-5} based on validation performance.

**Code Availability.** The code is made available at `https://github.com/decisionintelligence/VoT`.

### A.5 Effciency analysis

As shown in the Table 7, the total latency is about 5.9s, which is mainly due to the Event-driven branch. LLM indeed brings additional cost. However, text analysis can bring abundant and valuable information which can help break the bottleneck of time series forecasting. That's why we need to investigate multimodal and additional cost brought by LLM is inevitable. But in most domains where multimodal is needed, the data is always sampled at a low frequency due to the fact that the collection of text needs time. And the time gap is enough for us to do the next value prediction. Therefore, second-level latency is acceptable. And when there is a high demand for real-time performance, we can use only the numerical branch, which is much faster and also effective.

Table 7: Inference Latency of Main Model Components

| Branch / Module | Inference Time (s) |
|---|---|
| Event-driven Branch | 5.9 |
| Numerical Branch | 0.002 |
| AFF | 0.0005 |

## B HISTORICAL IN-CONTEXT LEARNING DESIGN DESCRIPTION

### B.1 THE ABLATION OF HIC

We compare our HIC method with simpler retrieval-based approaches and presented the results in the Table 8. In the Table 8, "No Retrieval" represents the baseline case where no retrieval mechanism is used. "Retrieve TS" refers to the time series retrieval method, where the retrieval score is calculated as the cosine similarity between time series and the recalled data is time series. "Retrieve Summary" is the summary-based retrieval method, comparing the summary embedding similarity and recalling the summary. "Full HIC" indicates our complete HIC method.

As the results show, for the Climate dataset, the time series retrieval method slightly outperforms the "No Retrieval" baseline, but the improvement is marginal. On the SocialGood dataset, the time series retrieval method even leads to a performance degradation compared to the "No Retrieval" baseline. The summary-based retrieval method shows a more significant improvement over the time series retrieval method in terms of performance metrics. However, our complete HIC method achieves the best performance, which indicates that the performance improvement is robust and primarily attributed to the full HIC module.

Table 8: Ablation Comparison of HIC with Simplified Retrieval Methods

| Dataset | Method | MSE | MAE |
|---------|--------|-----|-----|
| Climate | No Retrieval | 1.151 | 0.877 |
| | Retrieve TS Only | 1.137 | 0.866 |
| | Retrieve Summary Only | 1.091 | 0.846 |
| | **Full HIC** | **1.078** | **0.840** |
| SocialGood | No Retrieval | 0.845 | 0.410 |
| | Retrieve TS Only | 0.853 | 0.421 |
| | Retrieve Summary Only | 0.822 | 0.395 |
| | **Full HIC** | **0.804** | **0.389** |

## B.2 ROBUSTNESS OF HIC WHEN RETRIEVING WITH LOW SIMILARITY

We conducted supplementary experiments to compare the results under different scenarios of similar example selection. Specifically, we retrieved the 10th - most similar and 100th - most similar examples to simulate the scenario where there are no close historical examples. The results are shown in the following Table 9

From these results, it can be observed that as the retrieval range of similar examples changes (from the 1st-most to the 100th-most similar example), and in comparison with the no-retrieval case (equivalent to disabling some functions of the HIC module), although the model performance fluctuates, it generally remains relatively stable. This further demonstrates the effectiveness and robustness of our design under different conditions.

Table 9: Model Robustness to Imperfect Retrieved Examples

| Dataset | Retrieved Example by Similarity Rank | MSE | MAE |
|---------|--------------------------------------|-----|-----|
| Energy | 1st-most Similar Example | **0.222** | **0.343** |
| | 10th-most Similar Example | 0.229 | 0.348 |
| | 100th-most Similar Example | 0.233 | 0.348 |
| | No Retrieval | 0.238 | 0.363 |
| SocialGood | 1st-most Similar Example | **0.804** | **0.389** |
| | 10th-most Similar Example | 0.807 | 0.408 |
| | 100th-most Similar Example | 0.833 | 0.434 |
| | No Retrieval | 0.865 | 0.410 |

## B.3 ROBUSTNESS TO NOISY/SPARSE TEXT DATA

We manually created relevant datasets. In our experiments, we introduced two types of modifications to simulate different real-world scenarios. For the sparse scenarios, we applied 10% and 20% masking to the text in the datasets. This included both discrete point masking (mask the text $\mathbf{T}_i^{ex}$) and continuous point masking (mask $\{\mathbf{T}_i^{ex}, \mathbf{T}_{i+1}^{ex}, ...\}$). To simulate noisy scenarios, we added 10% and 20 noise to the text data. The experimental results, presented in the Table 10, show that while the performance does decrease slightly, the decline is not substantial. Moreover, our method still outperforms single-branch methods. These results demonstrate the robustness of our approach to noisy or sparse exogenous text. However, in extreme cases where all data is noisy or there is no valid text information available, the decline may be significant. In practice, severely degraded text data can be flagged via quality checks, mitigating negative impacts.

Table 10: Robustness Test of the Model Against Noisy and Sparse Text Data

| Dataset | Condition | MSE | MAE |
|---------|-----------|-----|-----|
| Climate | Mask 10% (mask10) | 1.099 | 0.850 |
| | Mask 20% (mask20) | 1.109 | 0.854 |
| | Noise 10% (noise10) | 1.098 | 0.850 |
| | Noise 20% (noise20) | 1.103 | 0.854 |
| | **Our Method (Original)** | **1.078** | **0.840** |
| SocialGood | Mask 10% (mask10) | 0.822 | 0.448 |
| | Mask 20% (mask20) | 0.854 | 0.451 |
| | Noise 10% (noise10) | 0.836 | 0.412 |
| | Noise 20% (noise20) | 0.876 | 0.480 |
| | **Our Method (Original)** | **0.804** | **0.389** |

## C  Endogenous Text Alignment Design Description

### C.1  Comparison with parameter-matched alternatives

we use gated-residual, cross-attention and FiLM for representation alignment and compare with ETA. The results are in Table 11. From these results, it is clear that methods like gated-residual and FiLM and cross-attention have higher MSE and MAE than ETA. Which means the methods struggle to effectively align the two highly distinct representation spaces of time series and text.

Table 11: Ablation Study on Temporal Alignment Methods

| Dataset | Method | MSE | MAE |
|---------|--------|-----|-----|
| Climate | Gated-Residual | 1.230 | 0.912 |
| | Cross-Attention | 1.202 | 0.893 |
| | FiLM | 1.211 | 0.897 |
| | **ETA** | **1.092** | **0.848** |
| Energy | Gated-Residual | 0.248 | 0.360 |
| | Cross-Attention | 0.254 | 0.369 |
| | FiLM | 0.252 | 0.367 |
| | **ETA** | **0.232** | **0.350** |
| SocialGood | Gated-Residual | 0.893 | 0.431 |
| | Cross-Attention | 0.887 | 0.427 |
| | FiLM | 0.894 | 0.431 |
| | **ETA** | **0.841** | **0.410** |

## D  Adaptive Frequency Fusion Design Description

### D.1  Why Choose Learnable Weights Instead of Linear Methods

Non-linear methods always have better fit ability. But we use 6 learnable parameter in stead of most non-linear method here for following reason. We need the Event-driven branch to forecast unseen patterns that cannot be learned from historical time series data and correct the numerical prediction. The unseen patterns always cause distribution shift. And we visualized the OT values of these datasets as shown in the Figure 8. We observe obvious distribution shifts between the training data and the test data. Therefore, non-linear methods may overfit the patterns learned from the training data and generalize poorly on the test data. In contrast, AFF uses only 6 parameters and aims to reflect the influence of text on time series, which generalizes better. We compare the results and provide them in the table below. The results in Table 12 show our AFF performs better than the non-linear methods, which support our analysis.

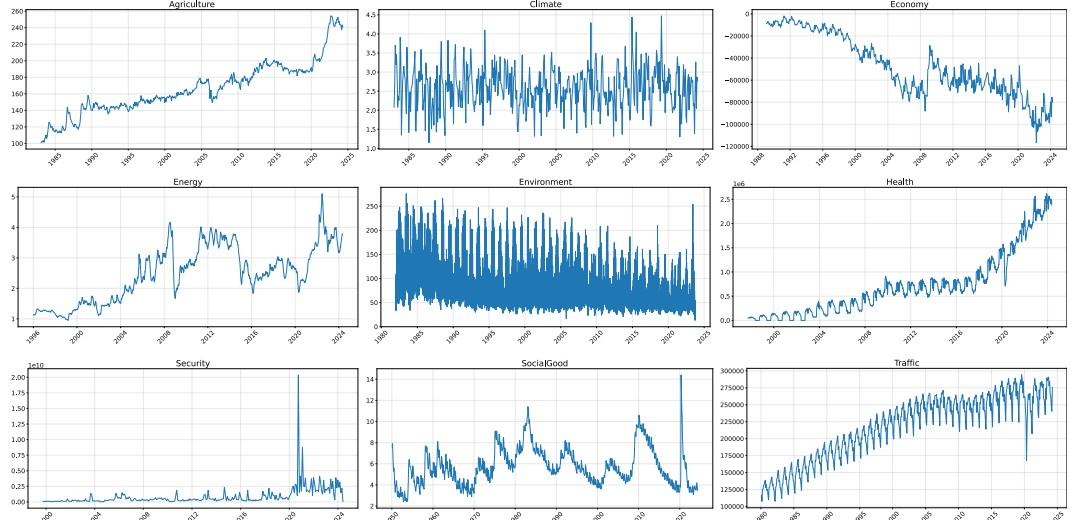

Figure 8: The visualization of the datasets

Table 12: Performance Comparison of Fusion Methods on Multiple Datasets

| Dataset | Fusion Method | MSE | MAE |
|---|---|---|---|
| Climate | MLP | 1.228 | 0.875 |
| | Cross-Attention | 1.768 | 1.055 |
| | **AFF** | **1.078** | **0.840** |
| Energy | MLP | 0.685 | 0.591 |
| | Cross-Attention | 0.430 | 0.502 |
| | **AFF** | **0.222** | **0.343** |
| SocialGood | MLP | 1.514 | 0.806 |
| | Cross-Attention | 1.459 | 0.842 |
| | **AFF** | **0.804** | **0.389** |

### D.2 EVENT AND NON-EVENT PART ANALYSIS

To distinguish the event part and the non-event part in the data, we visualized the test datasets of Energy and SocialGood dataset and visually identified event segments (sudden shifts) and non-event segments (stable periods) in the test sets. Subsequently, we computed the MSE and MAE of the predictions of the two periods from two branches (the event-driven branch and the numerical branch), as well as the fused final prediction, in comparison with the ground truth. The results are presented in the Table 13.

Upon analyzing these results, we observed that during the event period, the event-driven branch performs better than numerical predictions, and during the non-event period, numerical prediction is better. During the event period, the fusion method outperforms the numerical method, and during the non-event period, the fusion method is better than the event-driven method. This means the two predictions are complementary. For both the event part and the non-event part, the fusion method is better than both the event-driven and numerical prediction methods in some cases. For 'Overall', the fusion prediction is always better than the others, which is what we want. This can serve as proof that event-driven reasoning brings greater benefits to the event-related part than the harm to the non-event part, and ultimately leads to a better result.

We analyzed the Pearson coefficient between various frequency components and the ground truth to compare the correlation. The experiment is conducted on SocialGood dataset. The results are presented in the Table 14. As we previously mentioned in the paper, our event-driven predictions on low-frequency components tend to be closer to the ground truth, whereas numerical predictions on high-frequency components are closer to the ground truth.

Table 13: Prediction Performance Breakdown in Event and Non-Event Periods

| Period | Branch | Energy | | SocialGood | |
|---|---|---|---|---|---|
| | | MSE | MAE | MSE | MAE |
| **Event Part** | Event-driven Branch | 0.402 | 0.450 | **3.080** | **0.979** |
| | Numerical Branch | 0.443 | 0.523 | 3.339 | 1.022 |
| | Fusion Result | **0.362** | **0.460** | 3.084 | 0.969 |
| **Non-Event Part** | Event-driven Branch | 0.266 | 0.362 | 0.116 | 0.270 |
| | Numerical Branch | **0.105** | **0.257** | 0.068 | 0.208 |
| | Fusion Result | 0.125 | 0.269 | **0.066** | **0.205** |
| **Overall** | Event-driven Branch | 0.314 | 0.391 | 0.832 | 0.441 |
| | Numerical Branch | 0.244 | 0.360 | 0.861 | 0.405 |
| | Fusion Result | **0.222** | **0.343** | **0.804** | **0.389** |

Table 14: Pearson Correlation of Frequency Components (SocialGood)

| Pred Length | Comparison | High Freq | Low Freq |
|---|---|---|---|
| 8 | Event-driven vs Ground Truth | 0.155412 | 0.497763 |
| | Numerical vs Ground Truth | 0.423805 | 0.458464 |
| 10 | Event-driven vs Ground Truth | 0.265945 | 0.402431 |
| | Numerical vs Ground Truth | 0.349408 | 0.381150 |
| 12 | Event-driven vs Ground Truth | 0.273838 | 0.466686 |
| | Numerical vs Ground Truth | 0.310766 | 0.436835 |

# E    LLM GENERATION EXAMPLES FOR EVENT-DRIVEN PREDICTION

This section demonstrates the complete event-driven prediction pipeline through concrete examples from our US unemployment rate forecasting experiments.

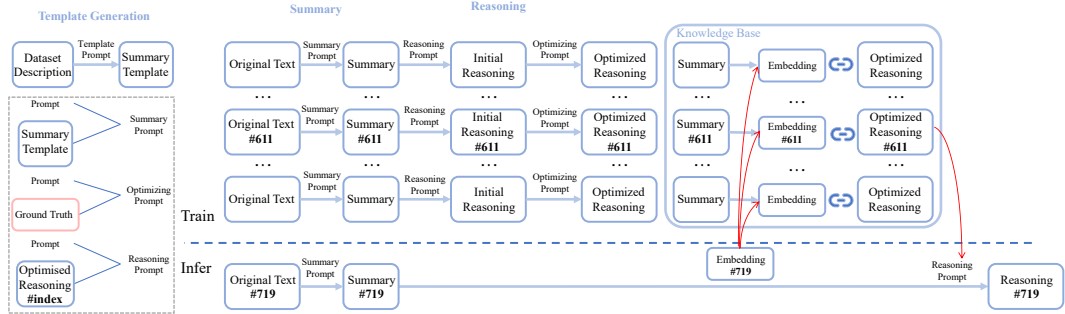

Figure 9: Illustration of the LLM generation pipeline for event-driven prediction. The process flows from template generation through summarization to reasoning, enhanced by Historical In-Context Learning that provides error-informed examples as guidance for improved forecasting accuracy.

## E.1    DATASET TEMPLATE GENERATION

- **Prompt Template:**

```
prompt = f"""You are a professional data scientist. Based on the first 10 records
{sample data pairs} and dataset description{dataset description}, generate a
summary template for OT value prediction with domain knowledge..."""
```

- **Generated Template (SocialGood Dataset):**

```
{
    "Dataset Name": "Unemployment Rate OT Value Prediction Dataset",
    "Description": "Monthly unemployment statistics for the United States,
disaggregated by race",
    "Data Type": "Time series data, text data",
    "Data Source": "U.S. Department of Labor, BLS",
    "Time Span": "1954 to present (monthly data), 1994 to present (annual
reports), 1979 to 1980 (text data)",
    "Sampling Method": "Comprehensive data from official reports",
    "OT Value": "Unemployment Rate",
    "Significance": "Indicator of economic health and societal impact",
    "Possible Relationships": {
      "Temporal": "OT value changes over time",
      "Spatial": "Racial disparities in unemployment rates",
      "Causal": "Economic policies and demographic changes"
    },
    "Features for Prediction": {
      "Trends in Time Series": "Historical trends and seasonality",
      "Key Information in Text Content": "Economic indicators and policy
changes",
      "Domain Knowledge": "Economics, labor markets, and demographic trends"
    },
    "Trend Analysis for Prediction": {
      "Short-Term Trends": "Recent fluctuations in the OT value",
      "Long-Term Trends": "Historical patterns and future projections",
      "Cyclical Trends": "Business cycle impact",
      "Demographic Trends": "Changes in population size and structure"
    }
}
```

## E.2 SUMMARY GENERATION

- **Prompt Template:**

```
base_prompt = f"""You are a professional data scientist analyzing a specific time
window of data.
TASK
Generate a NEW and UNIQUE analytical summary for this specific time window
(start_time) to {end_time}) of the dataset_name dataset.
REFERENCE INFORMATION (DO NOT COPY-ONLY FOR CONTEXT)
{dataset description}
REQUIRED OUTPUT STRUCTURE
Your output must follow this JSON structure, but with YOUR OWN analysis:
{template}
INPUT DATA TO ANALYZE
Time Series Data for this window:
{time_series_str}
Text Data for this window:
{texts_str}
"""
    instructions = """
IMPORTANT INSTRUCTIONS:
1. Your output must be a valid JSON object only.
2. DO NOT COPY descriptions from the reference information!
3. CREATE NEW analysis specific to THIS time window data.
4. Focus on the trends, patterns, and insights from THIS specific time period.
5. Your analysis should reflect what's happening in the provided time series and
text data.
Respond with ONLY the requested JSON object."""
```

- **Training Window #611 Summary (Dec 2000-Jul 2001):**

```
{
    "Dataset Name": "US Racial Unemployment OT Value Prediction Dataset",
    "Description": "Monthly unemployment statistics for the United States,
disaggregated by racial groups",
    "Data Type": "Time series data, text data",
    "Data Source": "U.S. Bureau of Labor Statistics (BLS)",
    "Time Span": "August 2001 to March 2002 (monthly data)",
    "Sampling Method": "Comprehensive data from official employment reports",
    "OT Value": "Average Unemployment Rate Across All Groups",
    "Significance": "Indicator of workforce dynamics and policy implications",
    "Possible Relationships": {
        "Temporal": "Seasonal fluctuations influenced by macroeconomic factors",
        "Spatial": "Regional differences impacted by industry composition and
skill requirements",
        "Causal": "GDP changes, monetary policy, and post-9/11 economic impacts"
    },
    "Features for Prediction": {
        "Trends in Time Series": "Historical seasonality and unemployment
patterns",
        "Key Information in Text Content": "Macroeconomic indicators and industry
composition changes",
        "Domain Knowledge": "Labor economics, racial demographics, and economic
shocks"
    },
    "Trend Analysis for Prediction": {
        "Short-Term Trends": "Overall downward trend during specified period (90%
confidence)",
        "Long-Term Trends": "Historical unemployment patterns across racial
groups",
        "Cyclical Trends": "Increased variability post-9/11 attacks (80%
confidence)",
        "Demographic Trends": "Variations in unemployment rates by racial group"
    },
    "OT": "[4.9, 4.7, 5.0, 5.3, 5.4, 6.3, 6.1, 6.1]"
}
```

- **Test Window #719 Summary (Dec 2009-Jul 2010):**

```
{
    "Dataset Name": "US Unemployment Trend Analysis",
    "Description": "Analyzing the trend of unemployment rate in the US from Dec
2009 to Jul 2010.",
    "Data Type": "Mixed time series and text data",
    "Data Source": "Bureau of Labor Statistics (BLS)",
    "Time Span": "Dec 2009-Jul 2010",
    "Sampling Method": "Regular reporting from multiple sources including
government agencies and academic institutions",
    "OT Value": "US Unemployment Rate",
    "Significance": "Indicates the overall health of the US economy and potential
impact on society",
    "Possible Relationships": {
      "Temporal": "Seasonal fluctuations and historical trends influencing
current unemployment rate",
      "Spatial": "Regional differences in unemployment rates indicating localized
economic conditions"
    },
    "Features for Prediction": {
      "Trends in Time Series": "Identifying short-term and medium-term trends
based on past performance",
      "Key Information in Text Content": "Understanding the underlying causes of
unemployment through contextual analysis",
      "Domain Knowledge": "Familiarity with macroeconomic factors affecting
employment rates"
    },
    "Trend Analysis for Prediction": {
      "Short-Term Trends": "Notable decline in unemployment rate after reaching
peak in Mar-Apr 2010 followed by slight rebound before stabilizing near initial
values",
      "Long-Term Trends": "Gradual upward movement since Sep-Dec 2009 indicates
stabilization of workforce participation",
      "Cyclical Trends": "Influence of business cycles contributing to periods of
accelerated hiring and layoffs throughout observation window",
      "Demographic Trends": "Shift toward part-time/full-time employment
distributions possibly linked to seasonal variations in service sector
activities"
    },
    "OT": "[9.7, 10.6, 10.4, 10.2, 9.5, 9.3, 9.6, 9.7]"
}
```

### E.3 INITIAL PREDICTION AND CORRECTED REASONING

- **Prompt Template:**

```
prompt = f"""[INST] You are a quantitative analyst specializing in multimodal
time series forecasting.
- Window Length: {window_size}
- Prediction Length: {prediction_length}
TASK: Predict next {prediction_length} values for {dataset_name} using textual
summary and window data.
ANALYSIS:
1. Textual Intelligence: Extract insights from summary
2. Numerical Patterns: Identify trends and cycles
3. Domain Knowledge: Apply sector understanding
EXAMPLE: {fixed_example}
SUMMARY: {summary}
OUTPUT: JSON format exactly:
```json
    "Prediction": "",
    "Reasoning": "Brief explanation of key factors driving the prediction"
```
[/INST]"""
```

- **Initial Prediction for Window #611:**

```
{
    "Summary..."
    "predicted_values": [6.0, 5.8, 5.6, 5.5, 5.4, 5.3, 5.2, 5.1, 5.0, 4.9, 4.8,
4.7],
    "actual_values": [5.7, 5.5, 6.0, 5.9, 5.7, 5.4, 5.3, 5.6, 5.7, 6.5, 6.4,
6.2],
    "original_reasoning": "The prediction reflects a gradual downward trend in
unemployment rates, consistent with the 90% confidence level for an overall
decline. The initial higher values account for post-9/11 variability (80%
confidence), followed by stabilization as macroeconomic recovery takes effect.
Seasonal adjustments and improving economic indicators suggest a return to
pre-shock levels over the 12-month horizon.",
    "ot_values": "[4.9, 4.7, 5.0, 5.3, 5.4, 6.3, 6.1, 6.1]"
}
```

- **Prompt Template:**

```
prompt = f"""[INST] You are an expert analyst tasked with improving prediction
reasoning by learning from actual outcomes.
DATASET: dataset_name
WINDOW SUMMARY: {summary}
PREDICTION ANALYSIS:
- Original Prediction: {Original Prediction}
- Actual Values: actual_str
ORIGINAL REASONING: {original_reasoning}
TASK: Rewrite the reasoning process by incorporating insights from the actual
outcomes. Focus on:
1. Pattern Recognition: What patterns in the actual data were missed or correctly
identified?
2. Factor Analysis: Which factors had stronger/weaker impact than initially
predicted?
3. Trend Understanding: How did the actual trend differ from expectations?
4. Domain Insights: What domain-specific knowledge can be extracted from this
comparison?
OUTPUT: JSON format exactly:
```json
{
    "Improved_Reasoning": "Comprehensive analysis incorporating actual outcomes
and improved understanding",
    "Key_Insights": "3-5 bullet points of key learnings from this prediction vs
reality comparison",
    "Prediction_Factors": "Updated understanding of the most important prediction
factors"
}
```
[/INST]"""
```

- **Corrected Reasoning:**

```
{
    "Summary..."
    "original_reasoning..."
    "improved_reasoning": "The original prediction anticipated a gradual
downward trend in unemployment rates, influenced by post-9/11 variability and
macroeconomic recovery. However, the actual data revealed a more complex pattern,
with fluctuations that did not align with the expected stabilization. Notably,
the actual unemployment rates showed heightened variability and an upward trend
toward the end of the period, contrary to the predicted decline. This discrepancy
suggests that the initial model underestimated the lingering effects of the 9/11
attacks and overestimated the speed of economic recovery. Incorporating these
insights, future predictions should account for prolonged external shocks and
their nonlinear impacts on unemployment trends.",
    "key_insights": [
      "The actual data exhibited more variability and an upward trend toward the
end of the period, contrary to the predicted gradual decline.",
      "The lingering effects of the 9/11 attacks had a more prolonged and severe
impact on unemployment rates than initially anticipated.",
      "Macroeconomic recovery was slower than predicted, highlighting the need
for more conservative estimates in post-crisis scenarios.",
      "Seasonal adjustments and industry composition changes may have been less
influential than external shocks in this period."
    ],
    "prediction_factors": "Updated understanding emphasizes the importance of:
1) accounting for prolonged external shocks (e.g., geopolitical events), 2)
adjusting recovery timelines based on real-time economic indicators, and 3)
incorporating nonlinear impacts of macroeconomic factors on unemployment rates."
}
```

### E.4   HIC-Guided Prediction (Test Phase)

- **RAG Retrieval for Window #719:**

```json
{
    "similar_example_window": 611,
    "similarity_score": 0.8278101086616516,
    "retrieval_method": "Summary_Based_RAG_Embedding"
}
```

- **Prompt Template:**

```
prompt = f"""[INST] You are a quantitative analyst specializing in multimodal
time series forecasting.
- Window Length: {window_size}
- Prediction Length: {prediction_length}
TASK: Predict next {prediction_length} values for {dataset_name} using textual
summary and window data.
ANALYSIS:
1. Textual Intelligence: Extract insights from summary
2. Numerical Patterns: Identify trends and cycles
3. Domain Knowledge: Apply sector understanding
EXAMPLE: {retrieved_example}
SUMMARY: {summary}
OUTPUT: JSON format exactly:
```json
    "Prediction": "",
    "Reasoning": "Brief explanation of key factors driving the prediction"
```
[/INST]"""
```

- **Generated Prediction with HIC:**

```json
{
  "prediction": [9.8, 9.9, 10.1, 10.2, 10.0, 9.8,
                 9.7, 9.6, 9.5, 9.4, 9.3, 9.2],
  "reasoning": "Analysis suggests gradual stabilization following
    pattern from similar historical window. Incorporates insights
    about underestimating short-term fluctuations. Prediction reflects
    balanced view considering cyclical nature and potential volatility
    learned from Window #611's error patterns."
}
```

### E.5   Analysis of Event-Driven Design

The above examples vividly illustrate how our event-driven prediction approach transforms the challenge of multimodal time series forecasting into a continuous learning process. Consider the journey of Window #611 during the 2000-2001 economic downturn: while the initial prediction correctly identified an upward trend in unemployment, it projected a smooth, gradual increase from 4.8% to 5.9%. Reality proved far more turbulent—unemployment spiked sharply to 6.3% following the September 11 attacks before showing unexpected resilience in recovery. This discrepancy between prediction and reality became a valuable learning opportunity. Through error analysis, the system discovered that external shocks create discontinuities that pure trend analysis cannot capture, leading to a fundamental recalibration of prediction weights: historical trends were reduced from 40% to 30% importance while economic shock factors increased from 20% to 35%. This learned knowledge proves its worth when the system encounters Window #719 during the 2009-2010 financial crisis period. Through embedding similarity analysis, the system recognizes a kindred pattern between these two economic downturns, achieving a similarity score of 0.8278. This recognition triggers the retrieval of Window #611's hard-won insights about volatility and recovery patterns. Rather than repeating the mistake of over-smooth predictions, the forecast for Window #719 now incorporates the understanding that crisis periods exhibit both sharp fluctuations and surprising stabilization mechanisms. The resulting prediction balances the observed stabilization trend with awareness of potential volatility, producing a more nuanced trajectory from 9.8% gradually declining to 9.2%. What makes this approach particularly compelling is how it mirrors human expert reasoning in economic forecasting. Just as economists draw parallels between historical crises to inform current predictions, our HIC systematically captures and transfers these insights across similar economic conditions. The system essentially builds a memory of how past events translated into numerical outcomes, creating a bridge between the rich semantic information in news reports, policy an-

nouncements, and economic analyses, and the quantitative requirements of time series forecasting. This design achieves what pure numerical models struggle with—understanding that a phrase like "unprecedented economic shock" carries specific implications for unemployment volatility based on historical precedent—while avoiding the computational expense of fine-tuning large language models for each specific domain.

## F ABLATION STUDY OF DIFFERENT REASONING MODELS

Table 15: Ablation study on different LLMs

| LLM | Energy | | Social Good | |
|---|---|---|---|---|
| | MSE | MAE | MSE | MAE |
| DeepSeek-R1 | 0.222 | 0.343 | 0.804 | 0.389 |
| GLM-4.5 | 0.229 | 0.337 | 0.829 | 0.411 |
| Qwen3-8B | 0.226 | 0.341 | 0.773 | 0.427 |

We also attempted to conduct experiments using different reasoning models. The experimental results showed in Table 15 indicated that under our method, LLMs could all provide excellent assistance for time series forecasting and outperform other methods.

## G COMPREHENSIVE EXPERIMENTAL RESULTS

This appendix provides comprehensive experimental results for all methods across different prediction horizons on 10 multimodal time series datasets.

Table 16: Model Performance Comparison Across Different Categories

| Category | | TS-only | | | | | | | | | | Text-enhanced | | | | | | | | Multimodal | | | | | | | |
| | | VoT | | numerical branch | | iTransformer | | PatchTST | | Raft | | iTransformer* | | PatchTST* | | Raft* | | GPT4TS | | GPT4MTS | | TATS | | TimeVLM | | CALF | |
| Metric | | MSE | MAE | MSE | MAE | MSE | MAE | MSE | MAE | MSE | MAE | MSE | MAE | MSE | MAE | MSE | MAE | MSE | MAE | MSE | MAE | MSE | MAE | MSE | MAE | MSE | MAE |
|---|---|---|---|---|---|---|---|---|---|---|---|---|---|---|---|---|---|---|---|---|---|---|---|---|---|---|---|
| Agriculture | 6 | 0.131 | 0.245 | 0.146 | 0.256 | 0.128 | 0.254 | 0.151 | 0.254 | 0.154 | 0.283 | 0.153 | 0.265 | 0.146 | 0.263 | 0.157 | 0.282 | 0.135 | 0.242 | 0.161 | 0.257 | 0.140 | 0.251 | 0.143 | 0.245 | 0.142 | 0.250 |
| | 8 | 0.182 | 0.286 | 0.188 | 0.287 | 0.200 | 0.296 | 0.197 | 0.287 | 0.194 | 0.297 | 0.195 | 0.296 | 0.189 | 0.310 | 0.237 | 0.344 | 0.198 | 0.284 | 0.207 | 0.288 | 0.187 | 0.282 | 0.215 | 0.287 | 0.195 | 0.285 |
| | 10 | 0.235 | 0.319 | 0.221 | 0.317 | 0.254 | 0.331 | 0.256 | 0.320 | 0.252 | 0.324 | 0.263 | 0.326 | 0.254 | 0.320 | 0.262 | 0.328 | 0.258 | 0.313 | 0.230 | 0.305 | 0.244 | 0.320 | 0.271 | 0.320 | 0.350 | 0.370 |
| | 12 | 0.288 | 0.355 | 0.301 | 0.356 | 0.300 | 0.352 | 0.309 | 0.351 | 0.301 | 0.383 | 0.305 | 0.352 | 0.338 | 0.369 | 0.326 | 0.379 | 0.291 | 0.338 | 0.301 | 0.342 | 0.290 | 0.350 | 0.322 | 0.359 | 0.314 | 0.355 |
| | avg | 0.209 | 0.302 | 0.214 | 0.304 | 0.220 | 0.308 | 0.228 | 0.303 | 0.226 | 0.322 | 0.229 | 0.310 | 0.232 | 0.316 | 0.246 | 0.333 | 0.220 | 0.294 | 0.225 | 0.298 | 0.215 | 0.301 | 0.238 | 0.303 | 0.250 | 0.315 |
| Climate | 6 | 1.071 | 0.840 | 1.090 | 0.847 | 1.111 | 0.860 | 1.197 | 0.896 | 1.298 | 0.925 | 1.117 | 0.857 | 1.206 | 0.898 | 1.292 | 0.924 | 1.207 | 0.901 | 1.199 | 0.895 | 1.194 | 0.897 | 1.218 | 0.907 | 1.231 | 0.910 |
| | 8 | 1.075 | 0.837 | 1.074 | 0.840 | 1.147 | 0.866 | 1.183 | 0.892 | 1.260 | 0.918 | 1.119 | 0.863 | 1.169 | 0.885 | 1.435 | 0.975 | 1.191 | 0.892 | 1.205 | 0.899 | 1.178 | 0.886 | 1.181 | 0.914 | 1.227 | 0.905 |
| | 10 | 1.078 | 0.836 | 1.087 | 0.843 | 1.152 | 0.870 | 1.188 | 0.886 | 1.301 | 0.932 | 1.118 | 0.858 | 1.168 | 0.882 | 1.296 | 0.929 | 1.169 | 0.886 | 1.173 | 0.885 | 1.170 | 0.881 | 1.179 | 0.880 | 1.508 | 0.990 |
| | 12 | 1.088 | 0.845 | 1.118 | 0.861 | 1.131 | 0.864 | 1.168 | 0.877 | 1.295 | 0.930 | 1.115 | 0.855 | 1.171 | 0.881 | 1.345 | 0.947 | 1.171 | 0.883 | 1.152 | 0.876 | 1.179 | 0.885 | 1.203 | 0.896 | 1.177 | 0.883 |
| | avg | 1.078 | 0.840 | 1.092 | 0.848 | 1.135 | 0.865 | 1.184 | 0.888 | 1.289 | 0.926 | 1.117 | 0.858 | 1.178 | 0.887 | 1.342 | 0.944 | 1.184 | 0.891 | 1.182 | 0.889 | 1.180 | 0.887 | 1.195 | 0.899 | 1.286 | 0.922 |
| Economy | 6 | 0.176 | 0.331 | 0.181 | 0.338 | 0.191 | 0.349 | 0.191 | 0.344 | 0.253 | 0.399 | 0.199 | 0.354 | 0.199 | 0.350 | 0.252 | 0.399 | 0.189 | 0.344 | 0.187 | 0.347 | 0.196 | 0.350 | 0.204 | 0.361 | 0.178 | 0.334 |
| | 8 | 0.195 | 0.346 | 0.193 | 0.349 | 0.204 | 0.360 | 0.203 | 0.356 | 0.253 | 0.402 | 0.208 | 0.363 | 0.216 | 0.367 | 0.279 | 0.425 | 0.213 | 0.370 | 0.202 | 0.357 | 0.214 | 0.366 | 0.223 | 0.379 | 0.232 | 0.377 |
| | 10 | 0.212 | 0.361 | 0.212 | 0.367 | 0.264 | 0.418 | 0.218 | 0.369 | 0.274 | 0.417 | 0.219 | 0.373 | 0.224 | 0.373 | 0.275 | 0.418 | 0.232 | 0.384 | 0.218 | 0.369 | 0.223 | 0.376 | 0.244 | 0.398 | 0.200 | 0.353 |
| | 12 | 0.223 | 0.374 | 0.227 | 0.377 | 0.230 | 0.385 | 0.229 | 0.382 | 0.280 | 0.425 | 0.227 | 0.379 | 0.239 | 0.388 | 0.294 | 0.437 | 0.233 | 0.386 | 0.226 | 0.379 | 0.229 | 0.381 | 0.245 | 0.396 | 0.216 | 0.363 |
| | avg | 0.201 | 0.353 | 0.203 | 0.358 | 0.222 | 0.378 | 0.210 | 0.363 | 0.265 | 0.411 | 0.213 | 0.367 | 0.219 | 0.370 | 0.275 | 0.420 | 0.217 | 0.371 | 0.208 | 0.363 | 0.215 | 0.368 | 0.229 | 0.384 | 0.207 | 0.357 |
| Energy | 12 | 0.091 | 0.218 | 0.101 | 0.227 | 0.121 | 0.258 | 0.107 | 0.235 | 0.123 | 0.255 | 0.124 | 0.271 | 0.105 | 0.232 | 0.116 | 0.242 | 0.111 | 0.243 | 0.111 | 0.244 | 0.105 | 0.232 | 0.114 | 0.253 | 0.102 | 0.224 |
| | 24 | 0.184 | 0.314 | 0.199 | 0.325 | 0.232 | 0.358 | 0.211 | 0.339 | 0.212 | 0.334 | 0.218 | 0.351 | 0.214 | 0.342 | 0.204 | 0.329 | 0.223 | 0.355 | 0.232 | 0.362 | 0.216 | 0.344 | 0.227 | 0.359 | 0.210 | 0.346 |
| | 36 | 0.278 | 0.393 | 0.300 | 0.406 | 0.279 | 0.399 | 0.298 | 0.405 | 0.306 | 0.413 | 0.321 | 0.423 | 0.308 | 0.413 | 0.292 | 0.402 | 0.314 | 0.423 | 0.308 | 0.418 | 0.309 | 0.418 | 0.309 | 0.410 | 0.300 | 0.420 |
| | 48 | 0.337 | 0.446 | 0.329 | 0.440 | 0.405 | 0.488 | 0.382 | 0.474 | 0.375 | 0.467 | 0.438 | 0.512 | 0.385 | 0.475 | 0.372 | 0.465 | 0.393 | 0.484 | 0.398 | 0.496 | 0.391 | 0.480 | 0.390 | 0.475 | 0.365 | 0.470 |
| | avg | 0.222 | 0.343 | 0.232 | 0.350 | 0.269 | 0.382 | 0.250 | 0.363 | 0.254 | 0.367 | 0.265 | 0.383 | 0.253 | 0.365 | 0.246 | 0.360 | 0.260 | 0.376 | 0.262 | 0.380 | 0.255 | 0.368 | 0.260 | 0.374 | 0.244 | 0.365 |
| Environment | 48 | 0.273 | 0.384 | 0.281 | 0.377 | 0.278 | 0.384 | 0.307 | 0.390 | 0.326 | 0.409 | 0.278 | 0.386 | 0.306 | 0.395 | 0.321 | 0.406 | 0.320 | 0.396 | 0.315 | 0.400 | 0.307 | 0.389 | 0.304 | 0.387 | 0.313 | 0.382 |
| | 96 | 0.270 | 0.378 | 0.273 | 0.381 | 0.281 | 0.387 | 0.329 | 0.400 | 0.321 | 0.413 | 0.284 | 0.393 | 0.333 | 0.406 | 0.317 | 0.410 | 0.340 | 0.401 | 0.340 | 0.401 | 0.334 | 0.402 | 0.327 | 0.405 | 0.335 | 0.394 |
| | 192 | 0.273 | 0.384 | 0.288 | 0.396 | 0.282 | 0.394 | 0.330 | 0.401 | 0.369 | 0.440 | 0.289 | 0.403 | 0.330 | 0.401 | 0.370 | 0.442 | 0.330 | 0.391 | 0.336 | 0.411 | 0.332 | 0.401 | 0.328 | 0.403 | 0.341 | 0.394 |
| | 336 | 0.256 | 0.374 | 0.256 | 0.374 | 0.264 | 0.379 | 0.301 | 0.391 | 0.342 | 0.428 | 0.263 | 0.378 | 0.300 | 0.385 | 0.341 | 0.429 | 0.300 | 0.383 | 0.299 | 0.390 | 0.302 | 0.391 | 0.320 | 0.395 | 0.312 | 0.377 |
| | avg | 0.268 | 0.380 | 0.274 | 0.382 | 0.276 | 0.386 | 0.317 | 0.395 | 0.339 | 0.423 | 0.278 | 0.390 | 0.318 | 0.397 | 0.337 | 0.422 | 0.322 | 0.393 | 0.323 | 0.400 | 0.319 | 0.396 | 0.320 | 0.398 | 0.325 | 0.387 |
| Health | 12 | 0.898 | 0.596 | 0.926 | 0.612 | 1.261 | 0.746 | 1.006 | 0.650 | 1.297 | 0.769 | 1.062 | 0.650 | 0.896 | 0.609 | 1.280 | 0.766 | 0.854 | 0.629 | 0.985 | 0.658 | 0.899 | 0.612 | 1.198 | 0.727 | 0.964 | 0.609 |
| | 24 | 1.210 | 0.707 | 1.275 | 0.729 | 1.476 | 0.816 | 1.403 | 0.799 | 1.773 | 0.963 | 1.936 | 1.025 | 1.326 | 0.766 | 1.792 | 0.973 | 1.271 | 0.751 | 1.513 | 0.802 | 1.307 | 0.759 | 1.491 | 0.839 | 1.451 | 0.749 |
| | 36 | 1.283 | 0.750 | 1.380 | 0.780 | 1.617 | 0.872 | 1.615 | 0.871 | 2.075 | 1.065 | 2.132 | 1.090 | 1.523 | 0.824 | 1.980 | 1.030 | 1.541 | 0.850 | 1.601 | 0.846 | 1.523 | 0.827 | 1.567 | 0.865 | 1.713 | 0.851 |
| | 48 | 1.429 | 0.802 | 1.459 | 0.810 | 1.723 | 0.896 | 1.706 | 0.897 | 2.186 | 1.104 | 1.721 | 0.894 | 1.694 | 0.871 | 2.100 | 1.081 | 1.698 | 0.878 | 1.757 | 0.889 | 1.693 | 0.872 | 1.702 | 0.907 | 1.836 | 0.889 |
| | avg | 1.205 | 0.714 | 1.260 | 0.733 | 1.519 | 0.833 | 1.432 | 0.804 | 1.833 | 0.975 | 1.713 | 0.915 | 1.360 | 0.768 | 1.788 | 0.963 | 1.341 | 0.777 | 1.464 | 0.799 | 1.356 | 0.767 | 1.490 | 0.835 | 1.491 | 0.775 |
| Security | 6 | 64.254 | 3.739 | 65.455 | 3.825 | 69.843 | 4.076 | 66.321 | 3.913 | 71.521 | 4.231 | 69.194 | 4.068 | 66.527 | 4.027 | 71.312 | 4.215 | 66.028 | 3.881 | 65.780 | 3.906 | 65.612 | 3.838 | 67.867 | 3.992 | 67.427 | 3.947 |
| | 8 | 68.175 | 3.838 | 68.608 | 3.951 | 72.230 | 4.114 | 69.892 | 3.951 | 74.775 | 4.398 | 72.014 | 4.083 | 69.938 | 4.093 | 73.483 | 4.387 | 69.279 | 4.033 | 68.914 | 3.955 | 71.860 | 4.146 | 70.928 | 4.084 | 69.608 | 3.993 |
| | 10 | 72.109 | 4.018 | 73.050 | 4.117 | 75.450 | 4.184 | 74.079 | 4.131 | 78.608 | 4.552 | 75.907 | 4.203 | 75.813 | 4.289 | 78.363 | 4.531 | 72.388 | 4.055 | 73.214 | 4.094 | 74.494 | 4.166 | 75.362 | 4.212 | 93.839 | 5.146 |
| | 12 | 75.931 | 4.153 | 75.686 | 4.078 | 82.644 | 4.493 | 77.815 | 4.254 | 83.911 | 4.712 | 79.013 | 4.261 | 78.607 | 4.299 | 83.189 | 4.659 | 76.965 | 4.219 | 78.041 | 4.316 | 77.656 | 4.239 | 80.767 | 4.438 | 74.631 | 4.113 |
| | avg | 70.117 | 3.937 | 70.700 | 3.993 | 75.042 | 4.217 | 72.027 | 4.062 | 77.204 | 4.473 | 74.032 | 4.154 | 72.721 | 4.177 | 76.587 | 4.448 | 71.165 | 4.047 | 71.487 | 4.068 | 72.406 | 4.097 | 73.731 | 4.182 | 76.376 | 4.300 |
| Social Good | 6 | 0.675 | 0.340 | 0.704 | 0.344 | 0.897 | 0.376 | 0.724 | 0.370 | 0.796 | 0.404 | 0.887 | 0.370 | 0.750 | 0.400 | 0.789 | 0.400 | 0.717 | 0.374 | 0.718 | 0.378 | 0.753 | 0.378 | 0.732 | 0.427 | 0.782 | 0.360 |
| | 8 | 0.774 | 0.375 | 0.806 | 0.395 | 0.897 | 0.431 | 0.829 | 0.416 | 0.907 | 0.463 | 0.887 | 0.426 | 0.848 | 0.411 | 0.940 | 0.444 | 0.855 | 0.459 | 0.942 | 0.505 | 0.875 | 0.409 | 0.822 | 0.427 | 0.874 | 0.386 |
| | 10 | 0.844 | 0.402 | 0.892 | 0.442 | 1.014 | 0.470 | 0.959 | 0.478 | 1.019 | 0.517 | 1.172 | 0.610 | 0.978 | 0.451 | 1.008 | 0.512 | 0.930 | 0.463 | 0.929 | 0.446 | 0.991 | 0.459 | 0.916 | 0.465 | 0.976 | 0.420 |
| | 12 | 0.925 | 0.438 | 0.962 | 0.457 | 1.153 | 0.574 | 1.263 | 0.643 | 1.151 | 0.553 | 1.285 | 0.653 | 1.059 | 0.477 | 1.141 | 0.550 | 1.167 | 0.608 | 1.093 | 0.470 | 1.053 | 0.474 | 1.005 | 0.505 | 0.991 | 0.439 |
| | avg | 0.804 | 0.389 | 0.841 | 0.410 | 0.961 | 0.463 | 0.944 | 0.475 | 0.968 | 0.484 | 1.027 | 0.515 | 0.909 | 0.427 | 0.970 | 0.477 | 0.917 | 0.476 | 0.920 | 0.450 | 0.918 | 0.428 | 0.869 | 0.444 | 0.906 | 0.401 |
| Traffic | 6 | 0.155 | 0.227 | 0.160 | 0.233 | 0.168 | 0.235 | 0.160 | 0.228 | 0.286 | 0.385 | 0.168 | 0.232 | 0.162 | 0.242 | 0.277 | 0.382 | 0.199 | 0.278 | 0.192 | 0.264 | 0.164 | 0.226 | 0.210 | 0.316 | 0.174 | 0.243 |
| | 8 | 0.167 | 0.231 | 0.170 | 0.236 | 0.176 | 0.232 | 0.171 | 0.230 | 0.274 | 0.372 | 0.183 | 0.241 | 0.168 | 0.228 | 0.325 | 0.418 | 0.204 | 0.262 | 0.195 | 0.256 | 0.178 | 0.242 | 0.212 | 0.313 | 0.176 | 0.232 |
| | 10 | 0.173 | 0.234 | 0.177 | 0.240 | 0.187 | 0.237 | 0.184 | 0.237 | 0.292 | 0.385 | 0.186 | 0.234 | 0.178 | 0.237 | 0.290 | 0.386 | 0.210 | 0.264 | 0.204 | 0.257 | 0.185 | 0.243 | 0.222 | 0.328 | 0.345 | 0.454 |
| | 12 | 0.181 | 0.239 | 0.181 | 0.241 | 0.205 | 0.250 | 0.188 | 0.239 | 0.300 | 0.387 | 0.198 | 0.242 | 0.188 | 0.246 | 0.308 | 0.392 | 0.211 | 0.268 | 0.218 | 0.268 | 0.189 | 0.242 | 0.222 | 0.322 | 0.193 | 0.243 |
| | avg | 0.169 | 0.232 | 0.172 | 0.237 | 0.184 | 0.238 | 0.176 | 0.234 | 0.288 | 0.382 | 0.184 | 0.237 | 0.174 | 0.239 | 0.300 | 0.394 | 0.206 | 0.266 | 0.203 | 0.261 | 0.179 | 0.238 | 0.217 | 0.320 | 0.222 | 0.293 |
| Weather | 48 | 0.861 | 0.662 | 0.843 | 0.655 | 1.273 | 0.833 | 0.894 | 0.658 | 0.922 | 0.675 | 0.904 | 0.658 | 0.888 | 0.651 | 0.920 | 0.672 | 0.902 | 0.650 | 0.897 | 0.666 | 0.901 | 0.650 | 0.885 | 0.656 | 0.957 | 0.669 |
| | 96 | 0.932 | 0.692 | 0.936 | 0.695 | 1.493 | 0.900 | 0.997 | 0.694 | 1.003 | 0.711 | 1.016 | 0.714 | 0.982 | 0.695 | 0.998 | 0.711 | 1.003 | 0.692 | 0.965 | 0.679 | 0.986 | 0.694 | 0.982 | 0.689 | 1.024 | 0.694 |
| | 192 | 0.998 | 0.710 | 1.010 | 0.722 | 1.048 | 0.733 | 1.057 | 0.723 | 1.124 | 0.760 | 1.044 | 0.732 | 1.033 | 0.715 | 1.120 | 0.759 | 1.066 | 0.721 | 1.036 | 0.751 | 1.054 | 0.717 | 1.097 | 0.731 | 1.111 | 0.721 |
| | 336 | 1.080 | 0.761 | 1.124 | 0.774 | 1.110 | 0.744 | 1.635 | 0.930 | 1.347 | 0.838 | 1.051 | 0.731 | 1.243 | 0.767 | 1.347 | 0.839 | 1.222 | 0.769 | 1.046 | 0.747 | 1.206 | 0.763 | 1.281 | 0.790 | 1.300 | 0.770 |
| | avg | 0.968 | 0.706 | 0.978 | 0.711 | 1.231 | 0.803 | 1.145 | 0.751 | 1.099 | 0.746 | 1.004 | 0.709 | 1.036 | 0.707 | 1.096 | 0.745 | 1.048 | 0.708 | 0.986 | 0.711 | 1.037 | 0.706 | 1.061 | 0.717 | 1.098 | 0.714 |
| | | 7.524 | 0.820 | 7.597 | 0.833 | 8.106 | 0.887 | 7.791 | 0.864 | 8.376 | 0.951 | 8.006 | 0.884 | 7.840 | 0.865 | 8.319 | 0.950 | 7.688 | 0.860 | 7.726 | 0.862 | 7.808 | 0.856 | 7.961 | 0.885 | 8.240 | 0.883 |

