# OpenReview forum: "Unlocking the Value of Text: Event-Driven Reasoning and Multi-Level Alignment for Time Series Forecasting"
_ICLR.cc/2026/Conference — ICLR 2026 Poster_

### Official Review · Reviewer_UX4Y · 2025-10-28

**Soundness:** 3
**Presentation:** 2
**Contribution:** 2
**Rating:** 4
**Confidence:** 3

**Summary:**

The paper presents VoT, a dual-branch framework for multimodal time-series forecasting. One branch performs event-driven reasoning over exogenous text through a three-stage pipeline enhanced by Historical In-Context Learning (HIC). The second branch conducts Endogenous Text Alignment (ETA) by converting statistics of the target series into textual descriptors and aligning them with numeric representations via contrastive learning. The two branches are combined using Adaptive Frequency Fusion (AFF), which learns frequency-band weights so that low-frequency, event-level effects and high-frequency fluctuations are fused coherently. Experiments on multiple datasets report consistent gains over strong baselines, with ablations attributing benefits to HIC, ETA, and AFF.

**Strengths:**

- Separating event-reasoning over exogenous text from numerical modelling, then fusing by learned frequency bands, is well motivated and clearly presented.
- The HIC retrieval strategy is a reasonable way to reuse corrected reasoning traces without fine-tuning.
- Strong empirical results span numerous datasets and include ablations that remove or modify key components.

**Weaknesses:**

- The generative pipeline plus HIC retrieval likely increases latency and cost; wall-clock measurements and throughput are not reported.
- The evaluation omits parameter-matched alternatives such as a gated residual or a lightweight cross-attention/FiLM plug-in, making it difficult to isolate architectural merit.
- Split protocols are not described in sufficient detail to exclude future-revealing or outcome-summarising language at time t.
- The stability of learned frequency weights under distribution shift and the sensitivity to band partitioning remain unclear.

**Questions:**

- What are wall-clock training and inference costs for the event pipeline and HIC retrieval, and how do they scale with text volume?
- How does VoT compare with (a) a small gated residual that can down-weight text channels and (b) a light cross-attention/FiLM block with similar parameter counts?
- How sensitive are results to the choice of frequency bands, and do learned fusion weights transfer across datasets or shifts?
- What is the behaviour when exogenous text is noisy, or sparse, and is there a fallback to ETA-only or TS-only prediction?

---

> ### Author Response · Authors · 2025-11-21
> **Response for W1 & Q1: Wall-clock time measurements**
>
> Thanks for your suggestion. As shown in our table, the total latency is  about 5.9s, which is mainly due to the Event-driven branch. LLM indeed brings additional cost. However, text analysis can bring abundant and valuable information which can help break the bottleneck of time series forecasting. That's why we need to investigate multimodal and additional cost brought by LLM is inevitable. But in most domains where multimodal is needed, the data is always sampled at a low frequency due to the fact that the collection of text needs time. And the time gap is enough for us to do the next value prediction. Therefore, second-level latency is acceptable. And when there is a high demand for real-time performance, we can use only the numerical branch, which is much faster and also effective. We have included this analysis **in Appendix A.5 of the updated revision**.
>
> | Branch / Module     | Inference Time (s) |
> |---------------------|--------------------|
> | Event-driven Branch | 5.9                |
> | Numerical Branch    | 0.002              |
> | AFF                 | 0.0005             |

---

> ### Author Response · Authors · 2025-11-21
> **Response for W2 & Q2: Comparison with parameter-matched alternatives**
>
> Thanks for your questions. The  experiments was carried out on the numerical branch. Here, we use gated-residual, cross-attention and FiLM for representation alignment and compare with ETA. The results are as follows:
>
> |  Dataset   |     Method      |  MSE  |  MAE  |
> | :--------: | :-------------: | :---: | :---: |
> |  Climate   | Gated-residual  | 1.230 | 0.912 |
> |            | Cross-attention | 1.202 | 0.893 |
> |            |      FiLM       | 1.211 | 0.897 |
> |            |       ETA       | 1.092 | 0.848 |
> |   Energy   | Gated-residual  | 0.248 | 0.360 |
> |            | Cross-attention | 0.254 | 0.369 |
> |            |      FiLM       | 0.252 | 0.367 |
> |            |       ETA       | 0.232 | 0.350 |
> | SocialGood | Gated-residual  | 0.893 | 0.431 |
> |            | Cross-attention | 0.887 | 0.427 |
> |            |      FiLM       | 0.894 | 0.431 |
> |            |       ETA       | 0.841 | 0.410 |
>
> From these results, it is clear that methods like gated-residual and FiLM and cross-attention have higher MSE and MAE than ETA. Which means the methods struggle to effectively align the two highly distinct representation spaces of time series and text. We have included this analysis **in Appendix C.2 of the updated revision**.

---

> ### Author Response · Authors · 2025-11-21
> **Response for W3: Dataset split and temporal leakage**
>
> Thanks for your question. Our dataset split is time-based. The first 70% of the data in temporal order serves as the training set, the 70%-80% segment serves as the validation set, and the latest 20% serves as the test set. There is no temporal overlap between the training, validation, and test sets. When constructing the knowledge base, we exclusively use relevant data from the training set. During the inference phase, that is, when the input is from the validation or test set, retrieving examples from the knowledge database only returns training-set data. All these data are temporally prior to the data in the validation and test sets, so no time leakage occurs. We have supplemented this explanation **in Appendix A.2 of the updated version**.

---

> ### Author Response · Authors · 2025-11-21
> **Response for W4 & Q3: Sensitivity and transferability of frequency weights**
>
> Thanks for your questions.
>
> - **The choice of frequency bands**
>
> We conducted tests under different boundary conditions to evaluate the sensitivity of the model and presented the results **in Figure 7** of the updated version. The figure displays a heat map of MSE or MAE values with respect to different low- and high-frequency boundaries. Through analysis of the results, we observe that with the change of low and high frequency bounds, the results do not change significantly when the boundary selection is reasonable. which means low sensitivity. The implementation details have been supplemented **in Appendix A.4 of the updated version**.
>
> And the fusion weight is learnable, so it doesn't need to be chosen; it's adaptive.
>
> - **The transferability of fusion weights**
>
> Different datasets will indeed learn different weights due to the adaptive mechanism. While the learned weights are dataset-specific and not directly transferable across vastly different domains, they can be effectively adapted to new, similar domains.
>
> We visualize the datasets **in Figure 8** of the updated version and observe obvious distribution shifts between the training data and the test data. It proves that our method is effective when distribution shift exists.

---

> ### Author Response · Authors · 2025-11-21
> **Response for Q4: Robustness to noisy/sparse text data**
>
> Thanks for your question. We manually created relevant datasets. In our experiments, we introduced two types of modifications to simulate different real-world scenarios. For the sparse scenarios, we applied 10% and 20% masking to the text in the datasets. This included both discrete point masking (mask the text $\mathbf{T}\_i^{ex}$) and continuous point masking (mask $\{\mathbf{T}\_i^{ex}, \mathbf{T}\_{i+1}^{ex},...\}$). To simulate noisy scenarios, we added 10% and 20% noise to the text data. The experimental results, presented **in the following table**, show that while the performance does decrease slightly, the decline is not substantial. Moreover, our method still outperforms single-branch methods. These results demonstrate the robustness of our approach to noisy or sparse exogenous text. However, in extreme cases where all data is noisy or there is no valid text information available, the decline may be significant. In practice, severely degraded text data can be flagged via quality checks, mitigating negative impacts. We have supplemented this explanation **in Appendix B.3 of the updated version**.
>
> |  Dataset   |    Condition    |  MSE  |  MAE  |
> | :--------: | :-------------: | :---: | :---: |
> |  Climate   |     mask10      | 1.099 | 0.850 |
> |            |     mask20      | 1.109 | 0.854 |
> |            |     noise10     | 1.098 | 0.850 |
> |            |     noise20     | 1.103 | 0.854 |
> |            | Ours (original) | 1.078 | 0.840 |
> | SocialGood |     mask10      | 0.822 | 0.448 |
> |            |     mask20      | 0.854 | 0.451 |
> |            |     noise10     | 0.836 | 0.412 |
> |            |     noise20     | 0.876 | 0.480 |
> |            | Ours (original) | 0.804 | 0.389 |

---

> ### Author Response · Authors · 2025-11-28
> **Looking forward to your feedback**
>
> Dear Reviewer UX4Y,
>
> We sincerely thank you for your time and the insightful comments on our paper. We have worked extensively to address the questions and concerns you raised.
>
> **As the discussion period is drawing to a close**, we hope our revisions have addressed your concerns and look forward to hearing from you. Please do not hesitate to contact us with any further questions or requests for clarification, and we are glad to provide prompt and detailed explanations as needed.
>
> All the best,
>
> Authors

---

### Official Review · Reviewer_jn6c · 2025-10-31

**Soundness:** 2
**Presentation:** 2
**Contribution:** 3
**Rating:** 4
**Confidence:** 4

**Summary:**

The paper proposes VoT (Unlocking the Value of Text), a framework that injects real-world textual context (e.g., news, reports, policies) into time-series forecasting through event-driven reasoning and multi-level alignment. VoT runs two prediction branches in parallel: (1) an event-driven LLM branch that converts external text into numeric forecasts via a three-stage pipeline (template → summary → reasoning) and a Historical In-Context (HIC) mechanism that retrieves past correction examples to reduce recurring errors; and (2) a numerical branch that aligns time-series embeddings with endogenous text describing trend/seasonality using ETA (representation-level alignment). Final predictions are fused in the frequency domain with AFF, which learns band-specific weights (low/mid/high) to exploit each branch’s strengths.
Contributions
- A principled event-to-forecast pipeline that turns text reasoning into numeric predictions, regularized by HIC.
- ETA: fine-grained representation alignment between TS embeddings and textual descriptions (trend/seasonality).
- AFF: prediction-level alignment via frequency-band fusion of event and numerical branches.
- Extensive benchmarks (10 domains) showing consistent gains over strong time-series and multimodal baselines, especially for event-sensitive regimes.

**Strengths:**

Originality: Proposes a dual-branch framework that combines Event-driven Reasoning from exogenous text with Multi-level Alignment (ETA for representation-level, AFF for prediction-level). The Historical In-Context Learning (HIC) idea—retrieving corrected past reasoning to guide current predictions—is a creative reuse of prior errors as in-context examples.

**Weaknesses:**

- The endogenous text appears to restate information already present in the series; the paper does not clearly justify why ETA (textualizing trend/seasonal components and aligning them) adds value beyond standard time–series decomposition and representation learning.
-  The computation of $Y_{\text{num}}$ is under-specified. It is unclear whether the forward pass at inference consumes only $H_{\text{ts}}$ or also the aligned textual features (e.g., $Z_{\mathrm{tr}}, Z_{\mathrm{se}}$).
- The construction and maintenance of the Knowledge Base $K$ are not operationalized: if an entry is stored per training window, $|K|$ may become very large. The paper does not detail pruning, refresh policies, or retrieval complexity/latency.

**Questions:**

- Could you add a baseline that performs ETA via simple TS–Text contrastive learning without trend/seasonal decomposition? Showing a consistent gap in favor of your decomposition-based ETA would strengthen the core claim.
- Why text-driven decomposition? Compared to standard time-series-based TS decomposition [1,2] applied to (H_{\text{ts}}), what concrete advantage does using endogenous text with TS–Text attention to produce (TS_Text_trend) and (TS_Text_seasonal) provide? Please quantify where text-guided components outperform classic decompositions.
- In Figure 6, the differences between the TS-only (pink) and event-driven (green) branches are not very clear across high- and low-frequency ranges. To make their roles more distinct, it might help to report a quantitative evaluation or show a case with a pronounced regime shift where the two branches diverge more visibly.

[1] Autoformer: Decomposition Transformers with Auto-Correlation for Long-Term Series Forecasting, NeurIPS21.
[2] Revitalizing Multivariate Time Series Forecasting: Learnable Decomposition with Inter-Series Dependencies and Intra-Series Variations Modeling, ICLR24.

---

> ### Author Response · Authors · 2025-11-21
> **Response for W1 & Q1 & Q2: The design of ETA**
>
> Thanks for your questions.
>
> - **Why endogenous text works**
>
> As you pointed out, endogenous text comes from the time series, and it seems to have no other information. However, inspired by Vision - Text multimodal methods, where the text serves as a caption and is highly similar to the picture in the semantic space, we also utilize the text as the caption of the time series to provide information. In fact, numerous time series multimodal methods, such as Time-LLM, employ endogenous text as a prompt to offer additional information. And we conducted the ablation experiments on the numerical branch. As shown in the Table, ETA outperforms the decomposition methods without modality alignment, indicating that it is indeed effective.
>
> - **Why decomposition works**
>
> To prove the effectiveness of decomposition and text-guide decomposition (ETA) , we compare the results in the Table. The results strongly support our core argument: across all three datasets (Climate, Energy, and SocialGood), ETA with decomposition consistently outperforms the method w/o decomposition. Notably, the decomposition we used is average pooling, which is same with Autoformer. The results demonstrates that integrating textual guidance into the decomposition and alignment process yields improvement beyond what pure numerical decomposition can achieve.
>
> We employ decomposition because it is a simple and widely-used technique in time series analysis. By decomposing the time series into trend and seasonal components, we can enrich the semantic representation of the time series to a certain degree. This enrichment allows for better alignment with textual information, facilitating more effective multi-modal interaction.  We have included this analysis **in Appendix C.1 of the updated revision**.
>
> |  Dataset   |   Fusion Method   |  MSE  |  MAE  |
> | :--------: | :---------------: | :---: | :---: |
> |  Climate   | w/o Decomposition | 1.120 | 0.859 |
> |            |  w/o TS-Text CL   | 1.184 | 0.888 |
> |            |        ETA        | 1.092 | 0.848 |
> |   Energy   | w/o Decomposition | 0.254 | 0.368 |
> |            |  w/o TS-Text CL   | 0.250 | 0.363 |
> |            |        ETA        | 0.232 | 0.350 |
> | SocialGood | w/o Decomposition | 0.892 | 0.440 |
> |            |  w/o TS-Text CL   | 0.944 | 0.475 |
> |            |        ETA        | 0.841 | 0.410 |

---

> ### Author Response · Authors · 2025-11-21
> **Response for W2: Clarification on $\mathbf{Y}_{num}$ computation**
>
> Thanks for your question. This issue is indeed an oversight on our part. The calculation formula of $\mathbf{Y}\_{num}$ is $\mathbf{Y}\_{num}=\frac{1}{2}\mathbf{H}^{\text{ts}} + \frac{1}{2}(\tilde{\mathbf{Z}}^{\text{tr}} + \tilde{\mathbf{Z}}^{\text{se}})$, which is obtained by fusing temporal and text representations. We have made the corresponding revisions **in section 3.4.1 of the updated revision**. Thank you again for pointing this out.

---

> ### Author Response · Authors · 2025-11-21
> **Response for W3: Knowledge base maintenance and scalability**
>
> Thanks for your question. The datasets we used are mostly small, and the retrieval complexity is O(k). When the data scale is too large, the maintenance of the knowledge base is necessary. However, our method can be simply combined with the existing methods for maintaining the knowledge base, such as saving the latest and most diverse data. Additionally, we provide results with different retrieval similarity and find that the performance still improves as the similarity decreases, which proves the robustness of knowledge against the loss of some examples caused by potential pruning.
>
>
> |  Dataset   |          Retrieval Based on Similarity Rank           |  MSE  |  MAE  |
> | :--------: | :---------------------------------------------------: | :---: | :---: |
> |   Energy   |               1st-most Similar Example                | 0.222 | 0.343 |
> |            |               10th-most Similar Example               | 0.229 | 0.348 |
> |            |              100th-most Similar Example               | 0.233 | 0.348 |
> |            | No Retrieval (Equivalent to no HIC-related Retrieval) | 0.238 | 0.363 |
> | SocialGood |               1st-most Similar Example                | 0.804 | 0.389 |
> |            |               10th-most Similar Example               | 0.807 | 0.408 |
> |            |              100th-most Similar Example               | 0.833 | 0.434 |
> |            | No Retrieval (Equivalent to no HIC-related Retrieval) | 0.865 | 0.410 |

---

> ### Author Response · Authors · 2025-11-21
> **Response for Q3: Quantitative analysis of frequency components**
>
> Thanks for your question. We analyzed the Pearson coefficient between various frequency components and the ground truth to compare the correlation. The experiment is conducted on SocialGood dataset. The results are presented  **in the following Table**. As we previously mentioned in the paper, our event-driven predictions on low-frequency components tend to be closer to the ground truth, whereas numerical predictions on high-frequency components are closer to the ground truth. We have included this analysis **in Appendix D.3 of the updated revision**.
>
> | pred length |                              | high freq | low freq |
> | :---------: | :--------------------------: | :-------: | :------: |
> |      8      | Event-driven vs Ground Truth | 0.155412  | 0.497763 |
> |             |  Numerical vs Ground Truth   | 0.423805  | 0.458464 |
> |     10      | Event-driven vs Ground Truth | 0.265945  | 0.402431 |
> |             |  Numerical vs Ground Truth   | 0.349408  | 0.38115  |
> |     12      | Event-driven vs Ground Truth | 0.273838  | 0.466686 |
> |             |  Numerical vs Ground Truth   | 0.310766  | 0.436835 |

---

> ### Author Response · Authors · 2025-11-28
> **Looking forward to your feedback**
>
> Dear Reviewer jn6c,
>
> We sincerely thank you for your time and the insightful comments on our paper. We have worked extensively to address the questions and concerns you raised.
>
> **As the discussion period is drawing to a close**, we hope our revisions have addressed your concerns and look forward to hearing from you. Please do not hesitate to contact us with any further questions or requests for clarification, and we are glad to provide prompt and detailed explanations as needed.
>
> All the best,
>
> Authors

---

### Official Review · Reviewer_Se43 · 2025-11-01

**Soundness:** 3
**Presentation:** 3
**Contribution:** 3
**Rating:** 6
**Confidence:** 3

**Summary:**

The paper introduces VoT (Value of Text), a multimodal time-series forecasting framework that integrates exogenous and endogenous text using a dual-branch architecture. The event-driven branch performs generative reasoning over textual events via Historical In-Context Learning (HIC), while the numerical branch aligns representations through Endogenous Text Alignment (ETA) and fuses predictions with Adaptive Frequency Fusion (AFF).

**Strengths:**

- Clear motivation that textual signals carry event-driven information complementary to numerical dynamics.
- Technically coherent pipeline combining LLM-based reasoning and representation alignment.

**Weaknesses:**

- The Event-Driven Reasoning and Historical In-Context Learning (HIC) components are interesting, but it remains somewhat unclear how much of the improvement truly comes from the reasoning process itself. Could the authors clarify whether HIC’s benefit persists when compared to simpler retrieval or embedding-based baselines? Also, since the reasoning pipeline involves curated summaries and correction prompts, how robust is it under domain shift or when applied without instruction-tuned LLMs?
- The averaged results suggest strong overall performance, but they do not reveal when text contributes most. Have the authors examined per-event or per-regime behavior—for instance, during shocks, sudden changes, or stable intervals? Such an analysis might clarify whether the textual reasoning module adapts dynamically to event intensity or merely improves overall fit.
- The Endogenous Text Alignment (ETA) and Adaptive Frequency Fusion (AFF) appear to rely on manually chosen decompositions (trend/seasonal, low/mid/high frequency). Are these partitions fixed, or are they optimized jointly with the forecasting backbone? If they are heuristic, have the authors tested sensitivity to these design choices or considered end-to-end learning variants?
- Given that the system depends on multiple LLMs and retrieval modules, can the authors discuss its computational footprint and inference latency? Is the framework feasible for continuous or real-time forecasting, or is it primarily an offline research prototype?

**Questions:**

- Can the authors provide a breakdown of performance during event vs. non-event intervals to demonstrate that “event-driven reasoning” indeed contributes selectively?
- How does HIC perform when no close historical correction exists (e.g., unseen event types)? Is retrieval robust to domain shift?

---

> ### Author Response · Authors · 2025-11-21
> **Response for W1 HIC's robustness**
>
> Thanks for your questions.
>
> - **Comparing with simpler retrieval methods**
>
> As you suggested, we have compared our HIC method with simpler retrieval-based approaches and presented the results **in the following table**. In the table, "No Retrieval" represents the baseline case where no retrieval mechanism is used. "Retrieve TS" refers to the time series retrieval method, where the retrieval score is calculated as the cosine similarity between time series and the recalled data is time series. "Retrieve Summary" is the summary-based retrieval method, comparing the summary embedding similarity and recalling the summary. "Full HIC" indicates our complete HIC method.
>
> | Dataset | Method | MSE | MAE |
> | :--: | :--: | :--: | :--: |
> | Climate | No Retrieval | 1.151 | 0.877 |
> |  | Retrieve TS | 1.137 | 0.866 |
> |  | Retrieve Summary | 1.091 | 0.846 |
> |  | Full HIC | 1.078 | 0.840 |
> | SocialGood | No Retrieval | 0.845 | 0.410 |
> |  | Retrieve TS | 0.853 | 0.421 |
> |  | Retrieve Summary | 0.822 | 0.395 |
> |  | Full HIC | 0.804 | 0.389 |
>
> As the results show, for the Climate dataset, the time series retrieval method slightly outperforms the "No Retrieval" baseline, but the improvement is marginal. On the SocialGood dataset, the time series retrieval method even leads to a performance degradation compared to the "No Retrieval" baseline. The summary-based retrieval method shows a more significant improvement over the time series retrieval method in terms of performance metrics. However, our complete HIC method achieves the best performance, which indicates that the performance improvement is robust and primarily attributed to the full HIC module. We have included this analysis **in Appendix B.1 of the updated revision**.
>
> - **The robustness against domain shift**
>
> Nowadays, LLMs have extensive domain knowledge. With HIC and a small amount of data, our method can be deployed across different domains and achieve good performance. For example, in our paper, we used the same LLM in 10 domains and observed improvements. This indicates that our method has good transferability and robustness to domains where LLMs have some knowledge. And in an extreme case where LLMs have no knowledge of a domain, we can use instruction fine-tuning along with our method to implement it. We have included this analysis **in Appendix B.2 of the updated revision**.

---

> ### Author Response · Authors · 2025-11-21
> **Response for W2 & Q1: Performance analysis on event vs. non-event periods**
>
> Thanks for your questions. To distinguish the event part and the non-event part in the data, we visualized the test datasets of Energy and SocialGood dataset and visually identified event segments (sudden shifts) and non-event segments (stable periods) in the test sets. Subsequently, we computed the MSE and MAE of the predictions of the two periods from two branches (the event-driven branch and the numerical branch), as well as the fused final prediction, in comparison with the ground truth. The results are presented **in the following table**:
>
> |                |              | Energy_MSE | Energy_MAE | SocialGood_MSE | SocialGood_MAE |
> | :------------: | :----------: | :--------: | :--------: | :------------: | :------------: |
> |   Event Part   | Event-driven |   0.402    |   0.450    |     3.080      |     0.979      |
> |                |  Numerical   |   0.443    |   0.523    |     3.339      |     1.022      |
> |                |    Fusion    |   0.362    |   0.460    |     3.084      |     0.969      |
> | Non-Event Part | Event-driven |   0.266    |   0.362    |     0.116      |     0.270      |
> |                |  Numerical   |   0.105    |   0.257    |     0.068      |     0.208      |
> |                |    Fusion    |   0.125    |   0.269    |     0.066      |     0.205      |
> |    Overall     | Event-driven |   0.314    |   0.391    |     0.832      |     0.441      |
> |                |  Numerical   |   0.244    |   0.360    |     0.861      |     0.405      |
> |                |    Fusion    |   0.222    |   0.343    |     0.804      |     0.389      |
>
> Upon analyzing these results, we observed that during the event period, the event-driven branch performs better than numerical predictions, and during the non-event period, numerical prediction is better. During the event period, the fusion method outperforms the numerical method, and during the non-event period, the fusion method is better than the event-driven method. This means the two predictions are complementary. For both the event part and the non-event part, the fusion method is better than both the event-driven and numerical prediction methods in some cases. For 'Overall', the fusion prediction is always better than the others, which is what we want. This can serve as proof that event-driven reasoning brings greater benefits to the event-related part than the harm to the non-event part, and ultimately leads to a better result. We have included this analysis **in Appendix D.2 of the updated revision**.

---

> ### Author Response · Authors · 2025-11-21
> **Response for W3: The sensitivity of design choice**
>
> Thanks for your questions.
>
> - **The design of ETA**
>
> The time series decomposition in the ETA component is grounded in a widely-used time series decomposition approach. We use average pooling for decomposition, which is a standard and parameter-free method. This process does not involve any manual selection.
>
> To further validate the effectiveness of this approach, we conducted supplementary ablation experiments on the numerical branch. The "w/o decomposition" condition indicates the absence of trend and seasonal decomposition, with only standard TS-Text contrastive learning being performed. The "w/o TS-Text CL" condition means that only time series decomposition is applied, without TS-Text contrastive learning. The experimental results, as presented in the following table, demonstrate that our proposed method, ETA, achieves superior performance compared to alternative designs. We have included this analysis **in Appendix C.1 of the updated revision**.
>
> |  Dataset   |   Fusion Method   |    MSE     |    MAE     |
> | :--------: | :---------------: | :--------: | :--------: |
> |  Climate   | w/o Decomposition |   1.120    |   0.859    |
> |            |  w/o TS-Text CL   |   1.184    |   0.888    |
> |            |        ETA        | 1.092  | 0.848  |
> |   Energy   | w/o Decomposition |   0.254    |   0.368    |
> |            |  w/o TS-Text CL   |   0.250    |   0.363    |
> |            |        ETA        | 0.232  | 0.350  |
> | SocialGood | w/o Decomposition |   0.892    |   0.440    |
> |            |  w/o TS-Text CL   |   0.944    |   0.475    |
> |            |        ETA        | 0.841  | 0.410 |
>
> - **Frequency bound choice**
>
> We conducted tests under different boundary conditions to evaluate the sensitivity of the model and presented the results **in Figure 7** of the updated version. The figure displays a heat map of MSE or MAE values with respect to different low- and high-frequency boundaries. Through analysis of the results, we observe that with the change of low and high frequency bounds, the results do not change significantly when the boundary selection is reasonable. which means low sensitivity. The implementation details have been supplemented **in Appendix A.4 of the updated version**.

---

> ### Author Response · Authors · 2025-11-21
> **Response for W4: Computational cost and inference latency**
>
> Thanks for your question. As shown in our table, the total latency is  about 5.9s, which is mainly due to the Event-driven branch. LLM indeed brings additional cost. However, text analysis can bring abundant and valuable information which can help break the bottleneck of time series forecasting. That's why we need to investigate multimodal and additional cost brought by LLM is inevitable. But in most domains where multimodal is needed, the data is always sampled at a low frequency due to the fact that the collection of text needs time. And the time gap is enough for us to do the next value prediction. Therefore, second-level latency is acceptable. And when there is a high demand for real-time performance, we can use only the numerical branch, which is much faster and also effective. We have included this analysis **in Appendix A.5 of the updated revision**.
>
> | Branch / Module     | Inference Time (s) |
> |---------------------|--------------------|
> | Event-driven Branch | 5.9                |
> | Numerical Branch    | 0.002              |
> | AFF                 | 0.0005             |

---

> ### Author Response · Authors · 2025-11-21
> **Response for Q2:  Robustness of HIC when retrieving with low similarity**
>
> Thanks for your question. We conducted supplementary experiments to compare the results under retrieving with different similarities. Specifically, we retrieved the 10th-most similar and 100th-most similar examples to simulate the scenario where there are no close historical examples. The results are shown **in the following Table**.
>
> |  Dataset   |          Retrieval Based on Similarity Rank           |  MSE  |  MAE  |
> | :--------: | :---------------------------------------------------: | :---: | :---: |
> |   Energy   |               1st-most Similar Example                | 0.222 | 0.343 |
> |            |               10th-most Similar Example               | 0.229 | 0.348 |
> |            |              100th-most Similar Example               | 0.233 | 0.348 |
> |            | No Retrieval | 0.238 | 0.363 |
> | SocialGood |               1st-most Similar Example                | 0.804 | 0.389 |
> |            |               10th-most Similar Example               | 0.807 | 0.408 |
> |            |              100th-most Similar Example               | 0.833 | 0.434 |
> |            | No Retrieval | 0.865 | 0.410 |
>
> From these results, it can be observed that as the similarity of examples decreases, and in comparison with the no-retrieval case (disabling the HIC module), although the model performance changes, it generally remains relatively stable. This further demonstrates the effectiveness and robustness of our design under different conditions. We have included this analysis **in Appendix B.2 of the updated revision**.

---

> ### Author Response · Authors · 2025-11-28
> **Looking forward to your feedback**
>
> Dear Reviewer Se43,
>
> We sincerely thank you for your time and the insightful comments on our paper. We have worked extensively to address the questions and concerns you raised.
>
> **As the discussion period is drawing to a close**, we hope our revisions have addressed your concerns and look forward to hearing from you. Please do not hesitate to contact us with any further questions or requests for clarification, and we are glad to provide prompt and detailed explanations as needed.
>
> All the best,
>
> Authors

---

### Official Review · Reviewer_Wk2i · 2025-11-02

**Soundness:** 3
**Presentation:** 4
**Contribution:** 3
**Rating:** 6
**Confidence:** 4

**Summary:**

The paper proposes VoT (Value of Text) a multimodal model to process text + numerical data for TSF. It uses an event-driven prediction branch and a numerical prediction branch to obtain branch wise predictions. event-driven prediction branch processes exogenous text data and uses retrieval to get relevant time-series data for historical in context learning (HIC). The numerical prediction branch aligns the time series with endogenous text that describes the time series. These branch wise predictions are then separated by 3 frequency bands (low, middle and high). The adaptive frequency fusion module (AFF) performs a linear combination at each band to then obtain to obtain the final forecast of the time series. The authors perform ablations of each module as well as perform experiments across 10 domains (from the MM-TSFLib benchmark + a new weather dataset) and 8 baselines to show superiority of the proposed method.

**Strengths:**

1. I like that the authors performed ablation on every module (ETA, AFF, HIC, Event branch), and appreciate the authors giving some insights rather than just presenting numbers. For instance, in section 4.3.1, the authors note that "removing HIC results in worse performance than removing the entire event-driven branch (w/o Event), suggesting that unguided LLM reasoning can be more detrimental than no reasoning at all.", which I think is a useful insight.

2. The pipeline of using LLMs to process exogenous text data to create summaries and time series, correcting the reasoning process to keep the correct samples and using a retrieval mechanism for historical in context learning seems a bit over-engineered. However, this is still an interesting idea, is novel (to the best of my knowledge), and helps improve performance.

3. The model achieves SOTA performance over the unimodal and multimodal baselines on all tasks.

**Weaknesses:**

1. The main weakness of the paper stems from the risk of temporal leakage in HIC module: It is crucial that the knowledge base never indexes summaries/corrections from future windows relative to the test horizon. The paper should spell out strict time-based splits for KB construction and retrieval, but HIC description lacks an explicit leakage guarantee.

2. The final prediction of the model is a linear combination of band-specific frequency components $\mathcal{F}^b_{*}$ between the exogenous (event-drive) and endogenous (numerical) branches. Why is this so? It seems quite reasonable to believe that the representation of these branches would have complex non-linear interactions. The paper would benefit from an ablation that models the final prediction as a nonlinear function instead (for example, MLP or cross attention).


Minor: The authors have provided a link to their code, but most files are not viewable (I get an error: "The requested file is not found").

**Questions:**

- How do the authors ensure in equation 7 that $E^{tr}$ and  $E^{se}$ extract trend and seasonality? What is enforcing this behaviour (the loss, prompt or something else)? This should be made clear in section 3.4.1. To me it does not seem like this is being enforced, in which case $Q^{tr}$ and $Q^{se}$ are essentially different query heads, capturing different information?

- Table 6 is quite large and bolding numbers would be beneficial to the reader.

 - Exactly how is the HIC knowledge base constructed per split? During inference at time t, are you guaranteed to retrieve only corrections derived from training windows strictly before t?

- How are the initial band boundaries chosen per dataset?

**Details Of Ethics Concerns:**

No ethical concerns

---

> ### Author Response · Authors · 2025-11-21
> **Response for W1 & Q3: Temporal leakage risk in HIC module**
>
> Thanks for your question. Our dataset split is time-based. The first 70% of the data in temporal order serves as the training set, the 70%-80% segment serves as the validation set, and the latest 20% serves as the test set. There is no temporal overlap between the training, validation, and test sets. When constructing the knowledge base, we exclusively use relevant data from the training set. During the inference phase, that is, when the input is from the validation or test set, retrieving examples from the knowledge database only returns training-set data. All these data are temporally prior to the data in the validation and test sets, so no time leakage occurs. We have supplemented this explanation **in Appendix A.2 of the updated version**.

---

> ### Author Response · Authors · 2025-11-21
> **Response for W2: Justification for linear fusion in AFF**
>
> Thank you for your suggestion to consider non-linear fusion methods. Your idea is indeed reasonable, and we have considered related designs during the development of our method. However, we identify certain limitations in these approaches. First,  we need the Event-driven branch to forecast unseen patterns that cannot be learned from historical time series data and correct the numerical prediction. The unseen patterns always cause distribution shift. We visualize the datasets **in Figure 8** of the updated version and observe obvious distribution shifts between the training data and the test data. Therefore, non-linear methods may overfit the patterns learned from the training data and generalize poorly on the test data. In contrast, AFF uses only 6 parameters and aims to reflect the influence of text on time series, which generalizes better. e compare the performance of AFF with non-linear fusion methods in the following table. The results show our AFF performs better than the non-linear methods, which supports our analysis. We have included this analysis **in Appendix D.1 of the updated revision**.
>
> |    Dataset     |  Fusion Method  |  MSE  |  MAE  |
> | :------------: | :-------------: | :---: | :---: |
> |  **Climate**   |       MLP       | 1.228 | 0.875 |
> |                | Cross-Attention | 1.768 | 1.055 |
> |                |       AFF       | 1.078 | 0.840 |
> |   **Energy**   |       MLP       | 0.685 | 0.591 |
> |                | Cross-Attention | 0.430 | 0.502 |
> |                |       AFF       | 0.222 | 0.343 |
> | **SocialGood** |       MLP       | 1.514 | 0.806 |
> |                | Cross-Attention | 1.459 | 0.842 |
> |                |       AFF       | 0.804 | 0.389 |

---

> ### Author Response · Authors · 2025-11-21
> **Response for Q1: How to ensure $\mathbf{E}^{tr}$ and $\mathbf{E}^{se}$ extract trend and seasonality**
>
> Thanks for your question. As you've pointed out, $\mathbf{Q}^{tr}$ and $\mathbf{Q}^{se}$ are indeed two distinct learnable query heads. We use these two learnable query to extract different types of information from the endogenous text. Subsequently, we perform cross-attention operations with the time series separately to obtain fused representations and we get $\bar{\mathbf{Z}}_i^{\text{tr}}, \bar{\mathbf{Z}}_i^{\text{se}}$. And then we carry out contrastive learning between $\bar{\mathbf{Z}}_i^{\text{tr}}, \bar{\mathbf{Z}}_i^{\text{se}}$ and the trend and seasonal components of the time series $\bar{\mathbf{H}}_i^{\text{tr}}, \bar{\mathbf{H}}_i^{\text{se}} \in \mathbb{R}^{d^{\text{ts}}}$ respectively. This approach effectively ensures that $\mathbf{Q}^{tr}$ and $\mathbf{Q}^{se}$ learn trend-related and seasonal information separately.
>
> We employ decomposition because it is a simple and widely-used technique in time series analysis. By decomposing the time series into trend and seasonal components, we can enrich the semantic representation of the time series to a certain degree. This enrichment allows for better alignment with textual information, facilitating more effective multi-modal interaction. To further validate the effectiveness of this approach, we conducted supplementary ablation experiments on the numerical branch. The "w/o decomposition" condition indicates the absence of trend and seasonal decomposition, with only standard TS-Text contrastive learning being performed. The "w/o TS-Text CL" condition means that only time series decomposition is applied, without TS-Text contrastive learning. The experimental results, as presented in the following table, demonstrate that our proposed method, ETA, achieves superior performance compared to alternative designs. We have included this analysis **in Appendix C.1 of the updated revision**.
>
> |  Dataset   |   Fusion Method   |    MSE     |    MAE     |
> | :--------: | :---------------: | :--------: | :--------: |
> |  Climate   | w/o Decomposition |   1.120    |   0.859    |
> |            |  w/o TS-Text CL   |   1.184    |   0.888    |
> |            |        ETA        | 1.092 | 0.848 |
> |   Energy   | w/o Decomposition |   0.254    |   0.368    |
> |            |  w/o TS-Text CL   |   0.250    |   0.363    |
> |            |        ETA        | 0.232  | 0.350  |
> | SocialGood | w/o Decomposition |   0.892    |   0.440    |
> |            |  w/o TS-Text CL   |   0.944    |   0.475    |
> |            |        ETA        | 0.841  | 0.410  |

---

> ### Author Response · Authors · 2025-11-21
> **Response for Q2: Table 6 formatting issue**
>
> Thank you for pointing out this issue. We have made corrections **in Table 6 (The Table 17 of the updated version)**.

---

> ### Author Response · Authors · 2025-11-21
> **Response for Q4: Selection of frequency band boundaries**
>
> Thanks for your question. **In Appendix A.3**, we defined the initial frequency bands as low frequency (0-10%), medium frequency (10-70%), and high frequency (70-100%).
>
> Additionally, we conducted tests under different boundary conditions to evaluate the sensitivity of the model and presented the results **in Figure 7** of the updated version. The figure displays a heat map of MSE or MAE values with respect to different low- and high-frequency boundaries. Through analysis of the results, we observe that with the change of low and high frequency bounds, the results do not change significantly when the boundary selection is reasonable. which means low sensitivity. The implementation details have been supplemented **in Appendix A.4 of the updated version**.

---

> ### Author Response · Authors · 2025-11-21
> **Response for Minor: The code repository**
>
> Thank you for pointing out the issue with the repository. We are resolving it and guarantee that it will be fixed during the rebuttal period.

---

> ### Author Response · Authors · 2025-11-28
> **Looking forward to your feedback**
>
> Dear Reviewer Wk2i,
>
> We sincerely thank you for your time and the insightful comments on our paper. We have worked extensively to address the questions and concerns you raised.
>
> **As the discussion period is drawing to a close**, we hope our revisions have addressed your concerns and look forward to hearing from you. Please do not hesitate to contact us with any further questions or requests for clarification, and we are glad to provide prompt and detailed explanations as needed.
>
> All the best,
>
> Authors

---

### Author Response · Authors · 2025-12-02
**Summary of Response**

Dear all,

We sincerely thank all reviewers, the PC, SAC, the original AC, and the new AC for your time and effort in coordinating and evaluating our submission. Due to the OpenReview incident and AC reassignment, we provide this summary to facilitate your understanding of our paper and the discussion during the rebuttal period.

### **Context of This Summary**

Following the incident, reviews were reverted to their pre-discussion state, and reviewers can no longer respond. While we provided comprehensive responses to every concern raised, we unfortunately did not receive response from reviewers before the discussion was frozen. However, we believe our responses systematically address all raised concerns supported by substantial additional experiments. This summary presents both the original concerns and our evidence-based responses for your evaluation.

### Reviewer Overview

| Reviewer | Rating |
| :------: | :----: |
|   Wk2i   |   6    |
|   Se43   |   6    |
|   jn6c   |   4    |
|   UX4Y   |   4    |

### Strengths

| Strengths | Reviewers |
|:----------|:------------------------:|
| **Innovative Design:** A novel dual-branch framework that integrates event-driven reasoning with numerical modeling, leveraging LLMs for textual inference and historical in-context learning. | Wk2i, jn6c, UX4Y |
| **Rigorous Execution:** Logically coherent integration of components and well-motivated module designs. | Se43, UX4Y |
| **Comprehensive Evaluation:** Achieved SOTA results backed by in-depth ablation studies across 10 datasets. | Wk2i, UX4Y |
| **Clear Motivation & Narrative:** Well-articulated premise that text provides complementary event context, with a clearly presented methodology. | Se43, jn6c, UX4Y |

### Response to Concerns

| Concerns | Reviewers | Response |
|:----------------:|-------------|:-----------------|
| Risk of temporal data leakage in HIC | Wk2i, jn6c | We have clarified the time-based split protocol (70%/10%/20% in chronological order) , ensuring no future data leakage. |
| HIC's robustness and effectiveness | UX4Y, jn6c | We simulated noisy and sparse text datasets and compared HIC with different retrieval methods. The results demonstrate HIC's robustness to imperfect data quality and its superior effectiveness over other retrieval baselines. |
| Justification for AFF vs. non‑linear methods | UX4Y | We compared AFF with non‑linear alternatives and analyzed why AFF was chosen and why it works effectively. |
| AFF frequency band choice and parameter sensitivity | Wk2i, Se43, jn6c | We conducted experiments and plotted a heatmap, which shows that the parameters are not sensitive within a reasonable range. |
| Effectiveness of ETA design (text-guided decomposition and alignment) | jn6c | We conducted ablation studies comparing ETA with variants that omit decomposition or contrastive learning. These confirm the effectiveness of the text‑guided decomposition and alignment design. |
| Comparison of ETA with other representation alignment methods | UX4Y | We compared ETA against other methods such as gated‑residual networks, showing that ETA achieves better performance in aligning time series and textual representations. |
| Performance on event vs. non‑event periods | Se43 | We manually identified event and non‑event periods in the test sets and compared predictions from the event‑driven and the numerical branch. The results confirm their complementarity—superiority of the event-driven branch in event periods and the numerical branch in non-event periods. |
| Quantify the differences between frequency components | jn6c | We compared the Pearson correlations between different frequency components and the ground truth, which confirms that each prediction branch aligns more closely with the ground truth on distinct frequency components. |
| Computational cost and inference latency | Se43, UX4Y | We provided an analysis of inference time. Inference time is longer due to LLM usage but remains viable for real-world deployment. |

Based on these responses, we have also revised the paper. We believe that our revisions have thoroughly addressed all reviewer concerns, and we are confident that the paper in its current form makes a strong contribution to the field of multimodal time series forecasting.

Thank you for your time and consideration.

Sincerely,
The Authors

---

### Meta-Review · Area_Chair_TT7j · 2025-12-16

**Summary:**

This paper proposes VoT, a dual-branch multimodal framework for time series forecasting that integrates exogenous and endogenous text via event-driven reasoning and multi-level alignment. The event-driven branch leverages LLM-based reasoning guided by Historical In-Context Learning (HIC), while the numerical branch performs representation-level alignment using Endogenous Text Alignment (ETA). Predictions from the two branches are fused in the frequency domain via Adaptive Frequency Fusion (AFF).

Reviewers generally agreed that the paper is well motivated, technically coherent, and empirically strong, with extensive experiments across 10 domains and comprehensive ablations. The primary concerns centered on (i) potential temporal leakage in HIC, (ii) the justification and robustness of the proposed alignment and fusion mechanisms relative to simpler or parameter-matched alternatives, (iii) the interpretability and necessity of endogenous text alignment, and (iv) computational cost and practicality, especially due to LLM usage.

The authors’ rebuttal was thorough and supported by substantial additional experiments and clarifications. In particular, they provided clear guarantees against temporal leakage, extensive ablations against alternative designs (e.g., gated residuals, cross-attention, FiLM, nonlinear fusion), sensitivity analyses, robustness studies under noisy/sparse text, and latency measurements. Overall, the rebuttal substantially strengthens confidence in the soundness and contribution of the proposed method.

**Reviewer Concerns:**

**Concerns adequately addressed:**

- **Temporal leakage in HIC (Wk2i, UX4Y):**
The authors clearly clarified the strict chronological split (70/10/20) and ensured that the HIC knowledge base is constructed exclusively from training data, with retrieval restricted to temporally prior windows. This concern is convincingly resolved.

- **Effectiveness and robustness of HIC vs. simpler retrieval (Se43, UX4Y):**
Additional comparisons against no-retrieval, time-series-only retrieval, and summary-based retrieval show consistent gains for full HIC. Further robustness tests under low-similarity retrieval and domain shift scenarios address the reviewers’ concerns.

- **Justification of AFF and linear fusion (Wk2i, UX4Y):**
The authors compared AFF against nonlinear alternatives (MLP, cross-attention) and provided a principled argument regarding generalization under distribution shift. Empirical results consistently favor AFF, adequately justifying the design choice.

- **Sensitivity to frequency band choices and stability (Wk2i, Se43, UX4Y):**
Heatmap-based sensitivity analyses demonstrate low sensitivity within reasonable ranges, alleviating concerns about heuristic partitioning.

- **Effectiveness of ETA and comparison to alternatives (jn6c, UX4Y):**
The authors added ablations against decomposition-free variants and parameter-matched alignment methods (gated residual, FiLM, cross-attention), consistently showing ETA’s superiority.

- **Event vs. non-event behavior (Se43, jn6c):**
Quantitative analyses confirm that the event-driven branch dominates during event periods, while the numerical branch performs better during stable regimes, validating the core motivation.

- **Computational cost and latency (Se43, UX4Y):**
Explicit inference-time breakdowns were provided, and the authors discussed practical deployment scenarios and fallbacks.

**Concerns partially outstanding or minor:**

- **Scalability of the HIC knowledge base (jn6c):**
While the authors discuss possible pruning strategies and robustness to reduced retrieval quality, large-scale or long-horizon deployments remain less thoroughly validated. This is a reasonable limitation but does not undermine the current contribution.

**Reviewer Scores:**

- **Reviewer Wk2i (initial: 6):**
Likely remains at 6, or slightly strengthened confidence within the same score, as the temporal leakage and fusion concerns were directly addressed with concrete evidence.

- **Reviewer Se43 (initial: 6):**
Likely remains at 6, with improved confidence due to added analyses on event vs. non-event regimes, robustness, and latency.

- **Reviewer jn6c (initial: 4):**
Likely increases to 6, as key concerns regarding ETA justification, ablations, operational details, and frequency analysis were thoroughly addressed.

- **Reviewer UX4Y (initial: 4):**
Likely increases to 6, given the added latency analysis, parameter-matched comparisons, sensitivity studies, and clarified split protocols.

---

### Decision · Program_Chairs · 2026-01-26

Accept (Poster)